



# Interpretable machine learning prediction of fire emissions and comparison with FireMIP process-based models

Sally S.-C. Wang[1], Yun Qian[1], L. Ruby Leung[1], Yang Zhang[2]

[1]Atmospheric Sciences and Global Change Division, Pacific Northwest National Laboratory, Richland, Washington, 99354, USA

[2] Department of Civil and Environmental Engineering, Northeastern University, Boston, Massachusetts, 02115, USA

*Correspondence to*: Sally S.-C. Wang (sing-chun.wang@pnnl.gov) and L. Ruby Leung (Ruby.Leung@pnnl.gov)

**Abstract.** Annual burned areas in the United States have increased twofold during the past decades. With more large fires resulting in more emissions of fine particulate matter, an accurate prediction of fire emissions is critical for quantifying the impacts of fires on air quality, human health, and climate. This study aims to construct a machine learning (ML) model with game-theory interpretation to predict monthly fire emissions over the contiguous US and to understand the controlling factors of fire emissions. By comparing the predicted fire $PM_{2.5}$ emissions from the interpretable ML model with the Global Fire Emissions Database (GFED) observations and predictions from process-based models in the Fire Modeling Intercomparison Project (FireMIP), the ML model is also used to diagnose the process-based models to inform future development. Results show promising performance for the ML model, Community Land Model (CLM), and Joint UK Land Environment Simulator-Interactive Fire And Emission Algorithm For Natural Environments (JULES-INFERNO) in reproducing the spatial distributions, seasonality, and interannual variability of fire emissions over CONUS. Regional analysis shows that only the ML model and CLM simulate the realistic interannual variability of fire emissions for most of the subregions (r>0.95 for ML and r=0.14~0.70 for CLM), except for Mediterranean California, where all the models perform poorly (r=0.74 for ML and r<0.30 for the FireMIP models). Regarding seasonality, most models capture the peak emission in July over western US. However, all models except for the ML model fail to reproduce the bimodal peaks in July and October over Mediterranean California, which may be explained by the coarse spatial resolutions of the processed-based models or atmospheric forcing data or limitations in model parameterizations for capturing the effects of Santa Ana winds on fire activity. Furthermore, most models struggle to capture the spring peak in emissions in southeastern US, probably due to underrepresentation of human effects and the influences of winter dryness on fires in the models. As for extreme events, both the ML model and CLM successfully reproduce the frequency map of extreme emission occurrence but overestimate the number of months with extremely large fire emissions. Comparing the fire $PM_{2.5}$ emissions from the interpretable ML model with process-based fire models highlights their strengths and uncertainties for regional analysis and prediction and provides useful insights on future directions for model improvements.

## 1. Introduction

        Large fires have increased across the United States over the past two decades, especially in the western US. While the total area burned in 2020 increased by 51% compared to the 10-year average for 2010-2019, the total number of fires in 2020 is smaller than the 10-year average. This indicates the contribution of larger and more powerful fires to the growing burned areas (NIFC, 2020). Large fires can directly lead to property damages and pose a threat to human lives (Thomas et al., 2017). Meanwhile, fine particulate matter (PM$_{2.5}$, particles with an





aerodynamic diameter smaller than and equal to 2.5 $\mu$m) emitted from fires not only have negative impacts on
human health but also affect climate and ecosystems (Johnston et al., 2012; Ward et al., 2012; Rap et al., 2013;
Kaulfus et al., 2017; Liu et al., 2018; Wang et al., 2018; Stowell et al., 2019). Driven by stronger fire heating and
with higher injection height, aerosols emitted from large fires can be transported to broader area and stay in
atmosphere longer. Given the increasing trend of fire emissions, fire smokes may become the predominant source
of $PM_{2.5}$ in the US in the future (Yue et al., 2013; Liu et al., 2016; Ford et al., 2018). Thus, an accurate prediction
of fire emissions is imperative for investigating the impacts of historical and future fires on air quality, human
health, and climate.
One of the widely used methods for predicting fire emission is process-based fire parameterization. These
process-based models generally employ universal functions depicting non-linear relationships between fires and
the input variables and apply the same functions to all grid cells in a model (Pechony and Shindell, 2009; Thonicke
et al., 2010). In addition, the parameters of the process-based model are usually determined by empirical or
statistical functions, assuming that the same parameters apply to all the regions or regions with limited fire
observations (Crevoisier et al., 2007; Parisien et al., 2016). Process-based models are usually included in the
dynamic global vegetation models (DGVMs) to simulate fire dynamics, vegetation dynamics, and biogeochemistry
driven by atmospheric forcing and socio-economic data  (Li et al., 2013; Knorr et al., 2016). Fire emissions,
including trace gases and aerosols, are calculated from the simulated fire carbon emissions and the emission factors,
with the former computed as the product of the burned area, fuel load, and combustion completeness. The process-
based models in DGVM coupled with other components of Earth system models can be used to assess the impacts
of environmental factors on fires and the feedback between fire emissions, land processes, and climate (Kloster et
al., 2010). In 2014, the Fire Model Intercomparison Project (FireMIP) was initiated to compare nine DGVMs that
include fire modules to better understand the performances of the global fire models (Rabin et al., 2017). The
FireMIP enables comprehensive evaluation and comparison across various process-based models and provides a
dataset of long-term fire simulations for regional and global analysis (Li et al., 2019; Hantson et al., 2020).
Besides process-based fire models, data-driven statistical models are also commonly used to estimate fire
activities using relationships between fires and predictor variables. Multiple linear regression (MLR) is a popular
simple statistical method used for fire modeling (Spracklen et al., 2009; Morton et al., 2013; Urbieta et al., 2015;
Williams et al., 2019). MLR can achieve a good performance, but it fails to capture the non-linear relationships
between fires and predictors, and it is sensitive to the collinearity and combinations of predictors (Littell et al.,
2009). Unlike MLR, machine learning (ML) is a novel tool for advancing fire modeling, given its strengths in
resolving the complex relationships between the target and predictor variables. Different ML approaches have been
used to estimate fire occurrence, burned areas, or emissions at various time scales and spatial scales (Cortez and
Morais, 2007; Aldersley et al., 2011; Dillon et al., 2011; Birch et al., 2015; Kane et al., 2015; Coffield et al., 2019;
Wang and Wang, 2020). Even though ML models generally achieve higher accuracy than simple statistical models,
their decision processes are often inscrutable, and hence lack interpretability. The development of explainable ML
represents major advances for scientific applications beyond predictions (Gunning, 2017; Barredo Arrieta et al.,
2020). For example, Wang et al. (2021) used the Extreme Gradient Boosting (XGBoost) algorithm and Shapley
Additive explanation (SHAP) to predict wildfire burned area and revealed the relationships between burned areas
and predictor variables. As process-based and data-driven models have their own advantages and weaknesses, as
listed in Table 1, comparing these models and assessing their uncertainties in historical simulations and future
projections are important. Yue et al. (2013) applied an MLR and a parameterization method to estimate burned
areas in ecoregions of the western US and found that both models explained ~50% of the variance in the observed
burned areas. Although they compared the burned areas estimated by the two methods and quantified their
uncertainties in fire projections, both methods are only driven by meteorology while the effects of fuels and human
activities are not considered.



The FireMIP dataset provides long-term simulations of multiple DGVMs with fire modules, allowing
comparisons between process-based and data-driven models, with all models considering all the potential factors
influencing fires, including climate, weather, vegetation, and human activities. This study aims to develop an ML
model with game theory interpretation for fire emission prediction and to understand controls of fire emissions.
The interpretable ML model is then used to reveal the important factors controlling fire emissions and diagnosis
the process-based FireMIP models. The ML model predicts the monthly $PM_{2.5}$ emissions from fires during 2000-
2020 at a spatial resolution of $0.25° \times 0.25°$ over the contiguous US (CONUS). It uses the XGBoost algorithm and
incorporates various predictors, including local and large-scale meteorology, land surface characteristics, and
socioeconomic variables, which are common input variables also used by the FireMIP models while some are
specifically related to fire activities in CONUS. The ML model and FireMIP models are optimized using different
data or predictors at various scales, which enables us to use the ML to diagnose the performance of FireMIP models
over CONUS through the comparisons of their performances and variable importance from the ML model. We
evaluate and compare the predicted fire emissions from the ML and FireMIP models against the GFED fire
emission product, focusing on spatial distributions, seasonality, and interannual variability over selected regions in
CONUS. Additionally, the ML model and the SHAP importance are used to identify the important drivers of fire
emissions in different regions and compare them with the corresponding parameterizations in the process-based
models. Lastly, we compare the process-based and ML model performances in simulating extremely large fire
emissions, including the spatial distributions of frequency and two case studies.
**2. Data**
**2.1 Fire-induced $PM_{2.5}$ emission data**
Monthly fire $PM_{2.5}$ emission data is obtained from the Global Fire Emissions Database (GFED). GFED
version 4 provides monthly burned area at 0.25º spatial resolution from 1997 to present, based on a combination of
the MODIS burned area product with active fire data from the Tropical Rainfall Measuring Mission (TRMM)
Visible and Infrared Scanner (VIRS) and Along-Track Scanning Radiometer (ATSR) family of sensors (Giglio et
al., 2013). The GFED fire $PM_{2.5}$ emissions are estimated by combining the burned area boosted by small fire burned
area (Randerson et al., 2012) and the emission factor data with a revised version of the Carnegie-Ames-Stanford
Approach (CASA) biogeochemical model that estimates fuel loads and combustion completeness for each monthly
time step (van der Werf et al., 2017). We use the GFED fire $PM_{2.5}$ emission as the target variable in the machine
learning model development and for model evaluation.
To reduce spatial heterogeneity and help model learning, we apply the inverse distance weighting (IDW)
(Bartier and Keller, 1996; Shepard, 1968) to interpolate the monthly gridded fire $PM_{2.5}$ emission at $0.25º \times 0.25º$.
The IDW method determines the value at a grid cell as the weighted average of the surrounding values within a
search distance, with the weights proportional to the inverse of the distance raised to the power value $p$. Here we
choose a value of 1 for $p$ and a search distance of 35 km for IDW processing. Note that the total fire emitted $PM_{2.5}$
within a search distance after IDW processing is constrained to be the same as the original data. In this study, we
only include grids with more than eight months of fire emissions larger than zero (in a total of 250 months),
encompassing 90% of the total fire emissions and ensuring sufficient data for the XGBoost model training. The
interpolated fire emission is normalized based on its 21-year mean and standard deviation for each grid to reduce
the skewness and improve data symmetry.



## 2.2 Predictor variables


We develop an empirical model at 0.25° × 0.25° grid resolution driven by various predictor variables at a
monthly scale from January 2000 to October 2020. Given the datasets have different spatial resolutions, all the
predictor variables are resampled to the spatial resolution of 0.25° × 0.25° by linear interpolation. The predictor
variables used in the model along with their original spatial and temporal resolutions are included in Table 2. Most
variables were also used in Wang et al. (2021) for developing an ML model of fire burned area over the contiguous
U.S.
*Local meteorology:* Same as the local meteorological predictors used in Wang et al. (2021), we include monthly
data of mean surface temperature, relative humidity (RH) at 2 m, daily precipitation, zonal (U) and meridional (V)
components of wind at 10 m from the North American Regional Reanalysis (NARR) (Mesinger et al., 2006) and
1000-hour dead fuel moisture (FM1000), Energy Release Component (ERC),  and vapor pressure deficit (VPD)
from the gridMET dataset (Abatzoglou and Kolden, 2013; Coffield et al., 2019). Drought is a natural phenomenon
that influences fires through ignition efficiency, fuel availability, and fuel moisture. Thus, we include the monthly
Standardized Precipitation Evapotranspiration Index (SPEI), a multiscalar drought index based on climatic data
(Vicente-Serrano et al., 2010). Given that lightning is one of the major ignition sources of fires and makes up
approximately 75% of burned areas in western US (Pyne, 1984; Stephens, 2005), in this study, we add the cloud-
to-ground (CG) lightning flash density from Severe Weather Data Inventory (SWDI) based on the National
Lightning Detection Network (NLDN) (Cummins and Murphy, 2009; NOAA, 2006). The daily number of CG
lightning flashes is summarized in 0.1° tiles and we aggregate the daily data to monthly scale.
*Large-scale meteorological patterns:* Large-scale meteorological patterns at a synoptic scale have been found to
link to large fire events (Crimmins, 2006; Trouet et al., 2009; Zhong et al., 2020; Dong et al., 2021). Furthermore,
it has been shown that including predictors of large-scale meteorological patterns conducive to wildfires
significantly improves the prediction of burned areas over CONUS (Wang et al., 2021). Thus, we follow the
methods developed by Wang et al. (2021) using the singular value decomposition (SVD) method to construct
predictors representing the synoptic patterns driving fire emission variability. Note that the only difference between
Wang et al. (2021) and this study is that they used wildfire burned area data and we use fire emissions to construct
the SVDs. Three regions where large fires periodically occur are selected for constructing SVDs: Northern
California, southern Rocky Mountains, and southeastern US, as defined in Wang et al. (2021). For each region, we
calculate the daily mean fire $PM_{2.5}$ emissions over the region and compute the day-to-day correlations between the
regional mean fire $PM_{2.5}$ emissions and the five gridded daily meteorological variables (surface temperature, 2-
meter RH, U-wind and V-wind at 850 hPa, and geopotential height at 500 hPa) for all 1° × 1° grid cells within the
large-scale domain, giving a correlation map for each meteorological variable. The correlation maps are then used
to derive the SVD modes representing the large-scale meteorological patterns related to fires. Finally, we compute
the monthly standard deviation of the daily SVD time series for the first two SVD modes, representing the month-
to-month variations of synoptic fluctuations and atmospheric instability. The detailed methods and discussions
about the SVDs are provided in Wang et al. (2021). Overall, the identified SVDs for the three regions are similar
to the SVDs in Wang et al. (2021) calculated using wildfire burned areas (Figs. S1-3).
*Land-surface properties:* We use the same set of variables in the burned area model that represent the effects of
fuel and land surface states on fire emissions, including evapotranspiration (ET), surface soil moisture, land types,
and topography (Wang et al., 2021). Monthly mean ET, vegetation fraction, and surface soil moisture are obtained
from the North American Land Data Assimilation System (NLDAS-2) (Xia et al., 2012). Land cover data of the
LAI classification scheme is obtained from the Terra and Aqua combined MODIS Land Cover Climate Modeling





Grid (CMG) Version 6 data (Friedl, 2015). Since the land cover data is at yearly intervals from 2001 to 2020, we
use the land cover data of 2001 for 2000. Topography data of slope and elevation is obtained from Amatulli et al.
(2018).

Besides the above-mentioned variables that were also used in Wang et al. (2021), in this study, we consider
the effect of fuel load on fire emissions, since fuel load is critical to fire emissions through its controls on fuel
consumption and burned areas (Parks et al., 2012; Liu and Wimberly, 2015). As there are limited observations of
fuel load, we use LAI to approximate the canopy bulk density and vegetation fraction to represent the existing
amount of vegetation. LAI is taken from MODerate resolution Imaging Spectroradiometer (MODIS) instruments
(Myneni et al., 2015) and vegetation fraction is obtained from the NLDAS-2. We also include fuel load simulated
by Community Land Model (CLM). Monthly fuel load data from 2000 to 2015 is obtained from a simulation by
CLM version 5 with biogeochemistry and prognostic crop, driven by atmospheric forcing from GSWP3v1
(Lawrence et al., 2019). The fuel load after 2015 is taken from a simulation under the SSP3 (shared socioeconomic
pathways) scenario. Additionally, we include normalized fuel load as a predictor to capture the effects of temporal
variation of fuel load, as the influence of fuel load on fire emissions is mainly attributed to its spatial variation
rather than the temporal variation (Lasslop and Kloster, 2015).

***Socioeconomic variables:*** We use population density and gross domestic product (GDP) per capita to represent
human effects on wildfires. The population density data is obtained from the Gridded Population of the World data
collection (GPW V4) for the years 2000, 2010, 2015, and 2020, with a spatial resolution of 30 arc-second (CIESIN-
Columbia University, 2017). The populations in other years are linearly interpolated between the abovementioned
four years. The GDP per capita is taken from a gridded global dataset for 2000-2015 with a spatial resolution of 5
arc minutes (Kummu et al., 2018). For the GDP after 2015, we use the data of 2015.
**3. Description of fire emission models**
**3.1 Process-based fire emission models**

The Fire Model Intercomparison Project (FireMIP) includes a set of common fire modeling experiments
from nine DGVMs driven by the same forcing data, allowing a better understanding of global fire models (Rabin
et al., 2017). The FireMIP dataset provides global gridded burned area fraction and fire emissions, including carbon
and 33 species of trace gases and aerosols over 1700-2012. Nine DGVMs with different fire modules are included
in FireMIP, including Community Land Model version 4.5 (CLM4.5) with the CLM5 fire module, Canadian
Terrestrial Ecosystem Model (CTEM), Jena Scheme for Biosphere-Atmosphere Coupling in Hamburg with Spread
and InTensity fire model (JSBACH-SPITFIRE; hereafter referred to as JSBACH), Joint UK Land Environment
Simulator with Interactive Fire And Emission Algorithm For Natural Environments (JULES-INFERNO; hereafter
referred to as JULES), Lund-Potsdam-Jena General Ecosystem Simulator with Global FIRe Model (LPJ-GUESS-
GlobFIRM; hereafter referred to as LPJ-Glob), LPJ-GUESS with SIMple FIRE model and Blaze-Induced Land-
Atmosphere Flux Estimator (LPJ-GUESS-SIMFIRE-BLAZE; hereafter referred to as LPJ-SIM), LPJ-GUESS with
SPITFIRE model (LPJ-GUESS-SPITFIRE; hereafter referred to as LPJ-SPI), MC2, and Organizing Carbon
Hydrology In Dynamic Ecosystems with SPITFIRE model (ORCHIDEE-SPITFIRE; hereafter referred to as
ORCHIDEE) (Rabin et al., 2017).
The nine DGVMs in FireMIP are driven by the CRU-NCEP v5.3.2 atmospheric forcing data with a spatial
resolution of 0.5° and a 6-hourly temporal resolution (Wei et al., 2014; Rabin et al., 2017). Other forcing data,





including annual global atmospheric $CO_2$ concentration, land use and land cover, and population density from 1700
to 2012 is taken from various data sources (Klein Goldewijk et al., 2010; Hurtt et al., 2011; Le Quéré et al., 2014).
Monthly cloud-to-ground lightning frequency with a resolution of $0.5° \times 0.5°$ over 1901-2012 is calculated based
on the observed relationship between present-day lightning and convective available potential energy (CAPE)
anomalies (Pfeiffer et al., 2013). Fire emissions in FireMIP are calculated considering the fire carbon emissions
and vegetation characteristics based on the plant functional type (PFT) from the FireMIP historical transient control
run (SF1). SF1 breaks the simulation period into three phases: the spin-up phase in 1700, the transient phase in
1701-1900, and the transient phase in 1901-2012 (see the detailed descriptions and model settings in Rabin et al.,
2017, Li et al., 2019, and Hantsan et al., 2020). In the 1901-2012 transient phase, the models are driven by time-
varying atmospheric forcing, $CO_2$ concentration, LULCC, population density, and lightning data. Note that the
MC2 and CTEM runs start from 1901 and 1861, while the rest of the models start from 1700. As the spatial
resolutions of the FireMIP models are different, the regridded model outputs with $1° \times 1°$ resolution obtained from
Li et al. (2019) are used to compare with the GFED data and the ML model.

### 3.2 ML-based approach: An eXtreme Gradient Boosting (XGBoost) model

The eXtreme Gradient Boosting (XGBoost) is a decision-tree-based ensemble machine learning method
using the gradient boosting approach (Chen and Guestrin, 2016). The XGBoost model builds multiple decision
trees that are added subsequently and learn the errors of the previous tree to reduce the loss and obtain the best
prediction. Unlike the gradient boosting machine (GBM) that also uses the gradient boosting approach, XGBoost
utilizes a more regularized model formalization to prevent over-fitting and improve the computational efficiency.
The formula for the prediction at step $t$ and grid location $i$ can be defined as follows:

$$\hat{y}_i^t = \sum_{k=1}^{t} f_k(x_i) = \hat{y}_i^{(t-1)} + f_t(x_i)$$

where $f_t(x_i)$ is the tree model at step $t$, $\hat{y}_i^t$ and $\hat{y}_i^{(t-1)}$ are the predictions at steps $t$ and $t-1$, and $x_i$ are the predictor
variables. The parameters of the model $f_t(x_i)$ are selected by optimizing the objective function that measures how
well the model fit the training data:

$$Obj^t = \sum_{i=1}^{n} L^t + \Omega^t$$

which is composed of the loss function $L^t$ and the regularizing term $\Omega^t$ in each step. $L_t$ is defined as $l(y_i, \hat{y}_i^{t-1} +$
$f_t(x_i))$ and $\Omega^t$ is defined as $\gamma T + \frac{1}{2}\lambda\|\omega\|^2$, where $\gamma$ is the regularization term which penalizes the number of
leaves in the tree $T$ and $\lambda$ is the regularization term which penalizes $\omega$, the weights of different leaves.
We use grid search to choose the set of suitable hyperparameters and achieve the best ML model
performance. Grid search is a tuning technique for computing the optimal values of hyperparameters considering
a range of numbers with a given increment. The parameter set that yields the best 5-fold cross-validation score is
selected as the final set of hyper-parameters. The considered hyper-parameters, their search domains, and the final
values are denoted in Table S1.
The 10-fold cross-validation (CV) technique is applied to evaluate the model and avoid overfitting. First,
we randomly divide the fire emission dataset (2000-2020 over CONUS) into ten equal-sized splits. Then, we train
the model with nine splits of the data and use the trained model to predict fire emissions for the remaining one split.





This process is repeated ten times for each split. Finally, the predictions are evaluated by grids and regions using
root mean square error (RMSE), correlation coefficient (R), and the index of agreement (IoA). The IoA represents
the ratio of the mean square error and the potential error, and the value closer to 1 indicates better agreement.
**3.3 Shapley additive explanations (SHAP)**
We utilize the SHAP to identify the relative importance of the predictor variables. SHAP is a novel
approach to resolve and explain variable importance based on game theory (Lundberg and Lee, 2017). Within the
scope of game theory, the goal is a prediction for a single observation. Each predictor variable is referred to as a
"player" in this game and contributes to the goal ("payout"). For each predictor, the SHAP variable importance
measures the marginal contribution considering all possible combinations of the predictor variables. The marginal
contribution is calculated by comparing the differences between the model fit $f_x(S \cup \{i\})$ including the predictor $i$
and another model fit $f_x(S)$ without predictor $i$. When there is more than one predictor $i$, the marginal contribution
also depends on the interactions with other predictors. Thus, the calculation repeats considering the whole set of
the predictors. The final contribution $\phi_i$ of predictor $i$ is the weighted average of all marginal contributions:
$$\phi_i = \sum_{S \subseteq F \setminus \{i\}} \frac{|S|! \, (F - |S| - 1)!}{F!} [f_x(S \cup \{i\}) - f_x(S)]$$

where $F$ is the total number of features, $S$ is the subset of predictors from all predictors except for predictor $i$,
$\frac{|S|!(F-|S|-1)!}{F!}$ is the weighting factor counting the number of permutations of the subset $S$. $f_x(S)$ is the expected
output given the predictors subset $S$. $[f_x(S \cup \{i\}) - f_x(S)]$ is the difference made by predictor $i$.
Compared to the commonly used feature importance, such as gain, or split count, SHAP is more consistent
and faithful to the model (Lundberg et al., 2019). More importantly, SHAP provides local importance that measures
the variable importance for each sample, while most of the feature importance metrics only have global importance
that measures variable contributions limited to the entire dataset. The global importance by SHAP is the average
of the absolute SHAP values for each predictor, providing an overall picture of the predominant variables
controlling fire emissions in CONUS. The local importance will be used to identify the important predictors for
large fire events in the ML model and diagnose the deficiency of the process-based models.
**4. Results**
**4.1 XGBoost model performance and variable importance**
Table 3 shows the whole CONUS and regional model performance, including RMSE, IoA, and correlation.
The model performs well at grid level over CONUS, with an RMSE of 0.16 g/m² and an IoA of 0.84. Figure 1a
shows the map of correlation between the observed and predicted monthly fire emission time series for each grid
over CONUS. Overall, the results indicate the ML model can reproduce the interannual variability of fire emissions
at 0.25° resolution over CONUS, with a mean correlation of 0.58 and more than 70% of the grids having
correlations larger than 0.4. To better assess model performance in different regions, Table 3 summarizes the model
performance for several selected regions: (1) western forest area, (2) Mediterranean California, (3) southwestern
US, and (4) southeastern US (color boxes in Fig. 1a). The regions where fires frequently occur are selected by the
similarity of ecoregions, vegetation types, and fire regimes. Figs. 1b-e show the time series of observed and
predicted fire PM₂.₅ emissions averaged over several regions. Generally, the ML model reproduces the interannual





variability of fire emissions for the selected regions (r=0.84-0.98). Among these regions, Mediterranean California
has the smallest correlation coefficient and largest RMSE compared to other regions, which can be explained by
the fact that fires in this region interact with multiple factors, including human activity, complex terrain, and Santa
Ana winds (Syphard et al., 2008; Yue et al., 2014). The interactions between fires and these factors pose
uncertainties and challenges in fire prediction over this region. It is also worth noting that the ML model captures
the large fire events in September 2020 in Oregon and California but underestimates the peak values by ~30%
(Figs. 1b and 1c).
To improve understanding of the ML prediction, we utilize the SHAP method to quantify the contributions
of each predictor variable to the prediction and identify the key contributing factors of fire $PM_{2.5}$ emission. Fig. 2
shows the 20 most important variables for the model ranked by the absolute mean SHAP values. Among the top
10 variables, seven of them are local meteorological variables, indicating local meteorology is the predominant
control of fire emissions, as these variables control fire activity directly (Liu and Wimberly, 2015; Abatzoglou et
al., 2016; Wang et al., 2021). Besides local meteorology, the predictors of large-scale meteorology (SVD1_SElag2
and SVD2_SElag2) are identified as the eighth and tenth important variables, showing that meteorology is not only
important at local scale but also at synoptic scale (Trouet et al., 2009; Pollina et al., 2013; Dong et al., 2021).
Finally, in addition to meteorology, fuel load is identified as the fifth important variable in the model, as fuel load
affects emission through controlling burned area and fuel consumption (Seiler and Crutzen, 1980). Considering the
important variables in different regions, the selected regions in western US (western forest area, Mediterranean
California, and southwestern US) generally share the common top 10 variables (Fig. S4). Over western US,
predictors controlling fuel dryness and fuel amount, including RH, fuel moisture (FM1000), ERC, vegetation
fraction, and fuel load, contribute more to fire emissions. On the other hand, large-scale meteorological patterns
(SVDs_SElag2) are more important for fire emissions in southeastern US.

## 298 4.2 General comparison between GFED, ML, and FireMIP models

This section compares the performance of the ML and FireMIP models benchmarked against observations
from GFED, and the evaluations are based on spatial distributions, seasonality, and interannual variability of fire
$PM_{2.5}$ emissions. Since the spatial resolutions of the GFED data, ML models, and FireMIP models are different,
they are all regridded to $1° \times 1°$ using bilinear interpolation. Note that the simulation period of FireMIP models
ends in 2012, so we use the overlapping period of 2000-2012 for comparison and exclude the MC2 model because
its simulation ends in 2008.

### 305 4.2.1 Spatial distributions of fire $PM_{2.5}$ emissions and sensitivities to RH and temperature

Fig. 3 compares the observed and simulated spatial distributions of long-term mean monthly fire $PM_{2.5}$
emissions averaged over 2000-2012. Among the models, the ML model, CLM, and JULES have better performance
in reproducing the spatial distributions of fire emissions over CONUS, with a correlation coefficient of 0.83, 0.52,
0.40, respectively. The ML model shows the best agreement with GFED, though it overestimates fire emissions
over Northern California. Both CLM and JULES simulate more $PM_{2.5}$ emissions over southeastern US, and JULES
overestimates fire emissions in Northern California. Some other models, such as CTEM, JSBACH, and LPJ-SIP,
tend to overestimate fire emissions over central US (e.g., Great Plains and Texas). LPJ-SIM captures the hotspots
of fire emissions over western US and southeastern US, but it simulates much more $PM_{2.5}$ emissions over the Rocky
Mountain and northeastern US. In terms of the total amount of $PM_{2.5}$ emissions, all models except ORCHIDEE-
SPITFIRE overestimate $PM_{2.5}$ emissions (8.33-79.49 Tg), compared to the GFED estimate of 4.98 Tg during 2000-
2012 over CONUS (Table 4).





The overestimations in some models may be explained by the sensitivities of fire emissions to individual
meteorological variables. Fig. 4 shows the slopes for the dependence of annual mean fire PM$_{2.5}$ emissions on annual
mean RH from the CRUNCEP atmospheric forcing data for GFED and the ten models based on linear regression.
Since the ML model uses NARR meteorology as predictors, we also include sensitivities of the fire emissions
predicted by the ML model to the NARR meteorology (Fig. 4b). Almost all models capture the negative dependence
of PM$_{2.5}$ emissions on RH over western US (r=-0.06~0.84), but the sensitivities in the models are much stronger
(steeper negative slope) than in GFED. For temperature, positive sensitivity is shown over western US in GFED
(Fig. 5), with the most significant slope in northern California. The sensitivities to temperature in models agree
with the observed sensitivities (r=-0.06~0.64), but some models show much stronger sensitivities over western,
central, and southeastern US. Generally, the spatial distributions of the long-term mean fire emissions shown in
Fig. 3 match well with the spatial distributions of sensitivities to RH or temperature, suggesting an important role
of the sensitivities in the model biases of predicting fire emissions. However, the correspondence of large fire
emissions to the sensitivities to RH or temperature shows regional differences. For instance, in western US, the
stronger sensitivities to both RH and temperature correspond to the overestimations in this region for most models,
including the ML model, CLM, CTEM, JSBACH, JULES, LPJ-SIM, and LPJ-SPI (Figs. 4 and 5). On the other
hand, over central US, larger PM$_{2.5}$ emissions simulated by CTEM and JSBACH only correspond to stronger
sensitivity to temperature (Fig. 5). Similar to central US, in southeastern US, the overestimations in CLM and
JULES only correlate with stronger sensitivity to temperature (Fig. 5). Regional differences in the correspondences
between the predicted fire emissions and their sensitivity to meteorology can be explained by several factors. For
western US, the overestimations of fire emissions correspond to both stronger sensitivities to RH and temperature,
given that fire activities are sensitive to fuel aridity that is controlled by temperature and fuel moisture (Abatzoglou
and Williams, 2016; Holden et al., 2018). As for southeastern US, fuels in this region typically burn at higher RH
and the interannual RH variation (standard deviation) is smaller (Balch et al., 2017; Brey et al., 2018). With higher
RH values and less variation in RH, the fire emissions in southeastern US show weaker sensitivity to RH than to
temperature in observation (Table S2). The above analysis shows that the overestimation of fire emissions in the
models may be attributed to the stronger sensitivities to meteorology. However, fire activities are controlled by
meteorology and other factors such as vegetation and human, so the analysis of fire emission sensitivity to
meteorology only provides a potential explanation to the overestimation of fire emissions in the models (Forkel et
al., 2019).
**4.2.2 Seasonality and interannual variability over CONUS**
In addition to evaluating spatial distributions, it is also important to compare the models' ability to
reproduce the temporal variability of fire emissions. As the models may systematically over-or underestimate fire
emissions, we normalize the emissions by the mean and standard deviation and focus only on its temporal
variability. Fig. 6a shows the seasonality of normalized fire PM$_{2.5}$ emission over CONUS. Most models capture the
seasonality of fire emission successfully (r>0.85), except LPJ-SIM which simulates peak emission in August-
October (r=0.65). Among the models, the ML model has the highest correlation coefficient between prediction and
observation from GFED (r=0.98) and successfully reproduces the peak in August. The seasonal peaks simulated
by the FireMIP models are broader and flatter than the peak in GFED, with an early peak in June-July continuing
to September (Fig. 6a).
In terms of interannual variability (Fig. 6b), the ML model, CLM, and JULES perform better than other
models, with larger correlation coefficient between simulated and observed fire PM$_{2.5}$ emissions (r=0.87, 0.71, and
0.55 for ML, CLM, and JULES, respectively; Table 4). Other models have relatively poor performance in capturing
the interannual variability. The interannual variability of fire emissions shows several peaks in 2002, 2007, and
2012 (black line in Fig. 6b), when western US contributes 76% of the total emissions to the peaks in these years.





Almost all models except ORCHIDEE capture the peak in 2012. However, most models miss the peaks in 2002
and 2007. Among all models, LPJ-Glob model simulates the peaks in the two years, while ML, JULES, and CLM
only capture the largest emission in 2007 (Fig. 6b).
**4.2.3 Seasonality and interannual variability by regions**

As the temporal variability of fire activities varies by region, we compare the performance between GFED
and the ML and FireMIP models by the regions defined in Sec. 4.1. Fig. 7 shows the seasonality and interannual
variability of normalized fire $PM_{2.5}$ emission over western forest area, Mediterranean California, southwestern US,
and the southeastern US. All models generally capture the seasonality of the western forest area peaking in summer,
with correlation coefficients larger than 0.8 (Table 4). Even though the FireMIP models generally reproduce the
peaks in summer, the predicted peaks are broad and flat, indicating a relatively longer fire season starting in June
and ending in September (Fig. 7a). When looking at the interannual variability, we find that the ML model has the
best performance with a correlation coefficient of 0.93, and it successfully captures the largest fire emission in
2007. CLM, JULES, and LGJ-Glob perform better than the rest of the models (r=0.70, 0.60, and 0.51 for CLM,
JULES, and LPJ-Glob, respectively; Table 4), but all of them still miss the peaks in 2007 and overestimate fire
emissions in 2001 and 2003 (Fig. 7b). The emission peak in 2007 is mainly attributed to the large fires in Idaho,
which were associated with synoptic weather patterns characterized by positive geopotential height and temperature
anomalies over the Pacific Coast and western US (Zhong et al., 2020). Consistent with prior findings, SHAP
importance shows that in the ML model SVD predictors (SVD_NCA and SVD_RM in July and August 2007 Fig.
8a) are the dominant factors of fire emissions in 2007 (contribute 27% and 28% for July and August 2007,
respectively), which are characterized by high pressure, low RH, and northeasterly winds over western US (Figs.
S1 and S2). Thus, the underestimation of peak emission in 2007 may be explained by the fact that the influences
of large-scale meteorology on fire activity are not fully considered in the FireMIP models, which are point models
driven only by local atmospheric forcing.
In Mediterranean California, the seasonality of fire emissions shows a bimodal pattern, peaking in August
and October. The peak in October is mainly due to the extremely large fires associated with Santa Ana winds in
2003 and 2007 (Keeley et al., 2009; Yue et al., 2014). The ML model simulates a flatter peak from July to October,
while all the FireMIP models except ORCHIDEE capture the first emission peak in summer but fail to simulate the
large fire emission in October (Fig. 7c). The underestimation associated with the Santa Ana winds is also shown in
the interannual time series in Fig. 7d. Several models, including LPJ-Glob, CTEM, LPJ-SPI, and JULES, capture
the peak in 2007 but only the ML model predicts both peaks in 2003 and 2007 even though the peak in 2003 is
underestimated. According to the SHAP importance from the ML model, the peak emissions in October 2003 and
October 2007 are mainly contributed by the SVD predictors and ERC (SVD2_NCA and SVD1_RM together
contribute 20% to the fire emissions for October 2003 and SVDs_SElag2 and SVD2_RM together contribute 31%
to the fire emissions for October 2007) and ERC (15% and 18% for October 2003 and 2007, respectively) (Fig.
8b). The results indicate that the ML model captures the effect of synoptic weather patterns on fire activity by
including the SVD predictors. Even though the wind speed is included in simulating fire spread in the FireMIP
models, the spatial resolutions of the models and/or the atmospheric forcing data may not be fine enough to resolve
the strengthened offshore winds through the complex terrain, and subsequently, they may not well capture the
effects of Santa Ana winds on fires. Besides the above-mentioned shortfall, all the models have problems
reproducing the interannual variability of the fire emissions over Mediterranean California, with very low
correlations (r<0.25) for the FireMIP models and a relatively low correlation (r=0.72) for the ML model (Table 4;
Fig. 7d). The poor performance for this region may be due to the complex relations between fires and multiple
factors, including meteorology, complex terrain, fuel, and human, which may not be fully represented in the models
(Mann et al., 2016; Radeloff et al., 2018).





Both the ML model and LPJ-SIM successfully reproduce the seasonality of fire emission in southwestern
US peaking in June (r=0.99 and 0.94 for ML and LPJ-SIM, respectively), while other models simulate relatively
smooth seasonality (Fig. 7e and Table 4). The ML model, LPJ-SIM, and ORCHIDEE have better performance for
the interannual variability, with correlation coefficients of 0.95, 0.40, and 0.45, respectively (Table 4). However,
most FireMIP models show larger variability in fire emissions than the GFED, and they all fail to capture the
extremely large fire emission in 2011 (Fig. 7f). The peak fire emission in 2011 over southwestern US was caused
by extremely low atmospheric moisture along with moderately high temperature, leading to record-breaking VPD
and wildfire activities (Williams et al., 2015). To explain why the FireMIP models fail to capture the peak of 2011,
we compare the VPD calculated from CRUNCEP data and the VPD data from gridMET used in the ML model. As
Fig. S5 shows, CRUNCEP shows smaller positive anomalies of VPD over southwestern US in 2011 summer, while
gridMET data demonstrates a significantly larger VPD anomaly. The biases in CRUNCEP data may partially
explain the underestimations in all FireMIP models.
For southeastern US, the seasonal cycle of fire $PM_{2.5}$ emissions displays a bimodal pattern, peaking in
spring (March-April) and fall (September and October) (Fig. 7g). Most models fail to reproduce the bimodal fire
emissions, but the ML model, LPJ-SIM, and LPJ-SPI can capture the bimodal pattern. Although LPJ-SIM and LPJ-
SPI predict the bimodal peaks, the first peak simulated by LPJ-SIM shows a one-month delay, and the second peak
simulated by LPJ-SIM and LPJ-SPI is one month early and one month late, respectively (Fig. 7g). In addition, the
ML model, CLM, and JSBACH reproduce the interannual variability of fire $PM_{2.5}$ emissions relatively well (r=0.96,
0.57, and 0.72 for the ML model, CLM, and JSBACH, respectively) (Table 4 and Fig. 7h). Interestingly, CLM and
JSBACH can capture several peaks in 2007, 2010, and 2011 but they do not simulate seasonality correctly, which
may be explained by the underestimation in spring compensated by the overestimations in summer related to
abnormal dryness or drought.

### 4.3 Performance in modeling extreme events

Fire activity in the US is becoming more hazardous, particularly over western US, due to more frequent
hotter and drier conditions as climate continues to warm (Williams et al., 2019). Thus, it is necessary to assess
whether the ML model and process-based models can capture the extreme events in terms of their magnitude,
frequency, timing, and location, which is essential to future projection and adaptation. As CLM performs relatively
well among the FireMIP models, we select CLM for comparison with the ML model at 1º × 1º resolution, focusing
on the spatial patterns of extreme event frequency and two case studies with extremely large fire emissions.

### 4.3.1 Frequency of extreme event occurrence

Fig. 9 shows the frequency maps of months with large fire emissions during 2000-2012 for GFED, the ML
model, and CLM. Large fire emission is defined as monthly fire $PM_{2.5}$ emissions greater than the 95[th] percentile of
fire $PM_{2.5}$ emission considering all the grids over CONUS in 2000-2012. GFED shows hot spots with a higher
frequency over northern California, the Pacific Northwest, and southeastern US, with total counts ranging from 15
to 105 (Fig. 9a). The ML model captures the spatial patterns (r=0.74), but it overestimates the number of months
by a factor of two to three compared to GFED, especially over western US (Fig. 9b). The spatial patterns of large
fire emission occurrence simulated by CLM are generally consistent with the observed distribution by GFED
(r=0.35). However, it overestimates the frequency, particularly over Idaho and northeastern US, and simulates more
significant numbers of months with extreme events over large spatial extents, may be due to its coarse spatial
resolution (Fig. 9c).



### 4.3.2 Case studies


To evaluate how well the models simulate the large fire emissions, we compare model performance for the
two recent cases reported to be the largest fire events during 2000-2012, including the fires in southern US in 2011
and western US in 2012. During 2011, a severe drought leading to large wildfires was observed over southern US,
including Arizona, New Mexico, and Texas (NOAA, 2012; Wang et al., 2015). Fig. 10 shows the maps of annual
mean fire $PM_{2.5}$ emissions over southern US from GFED, the ML model, and CLM. GFED shows the largest fire
emissions close to the border of Arizona and New Mexico in conjunction with other small hotspots over New
Mexico, Texas, and Louisiana (Fig. 10a). The ML model overall reproduces the spatial distributions of the fire
emissions (r=0.96) and captures the largest fire emission in Arizona and New Mexico in 2011 (Fig. 10b). However,
CLM does not capture the hotspots observed in GFED over Arizona and New Mexico but simulates larger fire
emissions in Louisiana instead (Fig. 10c). In terms of the time series, the ML model reproduces the temporal
variability of fire emissions and successfully captures the peak of total fire $PM_{2.5}$ emissions in June 2011 (r=0.98;
Figs. 10d and 10e). Although CLM simulates the peak in June, it overestimates fire emissions in the following
months by a factor of 4 (r=0.52; Figs. 10d and 10e).
In 2012, western US experienced several major wildfires (NOAA, 2013). The warm and dry conditions led
to large wildfires in California, Oregon, New Mexico, and Colorado (Fig. 11). Both the ML model and CLM
capture the hotspots with large fire emissions (Fig. 11b and 11c) and have correlation coefficients of 0.56 and 0.37,
respectively. However, the ML model tends to overestimate fire emissions, especially in areas surrounding the
grids with extremely large fire emissions (Fig. 11b). CLM misses some large fire emissions in Colorado and New
Mexico and underestimates the larger fire emissions in several hotspots (Fig. 11a), which may be explained by its
coarse resolution. The time series of normalized fire $PM_{2.5}$ emissions in 2012 show one peak in August. The ML
model captures the peak and presents a high correlation between the simulated and observed normalized and total
$PM_{2.5}$ fire emissions (r=0.98). CLM captures the peak in August but overestimates emissions in September and
October (r=0.84; Figs. 11d and 11e).

### 5. Discussion and conclusions


This study provides the first assessment to evaluate the performance of data-driven and process-based
models in predicting fire $PM_{2.5}$ emissions over CONUS. We first demonstrate that the developed ML model
performs well in predicting monthly fire $PM_{2.5}$ emissions nationwide at grid cells of 0.25º × 0.25 º resolution from
2000 to 2020, with an RMSE of 0.16 $g/m^2$ and IoA of 0.84. The ML model outperforms prior statistical models
predicting fire activities at similar spatial and temporal scales (Carvalho et al., 2008; Bedia et al., 2014).
Considering the performance at a regional scale, the ML model reproduces the interannual variability of fire
emissions for the selected regions, with correlation coefficients ranging from 0.84 to 0.98. Therefore, the ML model
has a promising performance in predicting fire emission over CONUS at a relatively fine spatial resolution.
Compared to the wildfire burned area model in Wang et al. (2021), the fire emission model in this study shows
slight degradations in capturing the interannual variability of fire emission at grid level (e.g., percentage of grids
with correlations larger than 0.4). This may be explained by the fact that the fire emission model may not effectively
resolve the relationships between fires and predictors when more grids with less fire occurrence are included (i.e.,
more zeros or unburned grids) without reliable information about ignition. As a side note, both burned area and
emission ML models have relatively poor performance over Mediterranean California, indicating the challenges in
modeling fire activities in this region where the terrain and land use are complex. The SHAP variable importance
shows that meteorology at both local and synoptic scale as well as fuel loads are important variables controlling
fire emissions over CONUS. Regional analysis of predictors indicates that fuel dryness such as fuel moisture and




energy release component (ERC) and fuel load are important for predicting fire emissions in western US, while
large-scale meteorological patterns (SVDs_SElag2) contribute more to fire emissions in southeastern US.
We then compare the simulated fire $PM_{2.5}$ emissions from the ML model and FireMIP models against
GFED from 2000 to 2012 at the spatial resolution of $1º × 1º$. The ML model, CLM, and JULES reproduce the
spatial distribution more reasonably than the rest of the FireMIP models (r=0.83, 0.52, and 0.40 for the ML, CLM,
and JULES, respectively). Both CLM and JULES simulate more fire $PM_{2.5}$ emissions over southeastern US, which
can be explained by several reasons. First, it has been shown that the satellite-observed burned areas in southeastern
US are much smaller than the burned areas estimated from the ground-based fire records, which might have resulted
from the small prescribed and agricultural fires (Hu et al., 2016; Nowell et al., 2018). In addition, large differences
exist among different satellite estimated fire $PM_{2.5}$ emissions in southeastern US (Li et al., 2019). As a consequence,
these studies highlighted uncertainties about the GFED estimated burned area and emission over southeastern US.
Second, cropland fires are one of the predominant fire types in this region. Among the FireMIP models, CLM is
the only model that simulates cropland fires (Li et al., 2013). For JULES, even though it does not simulate cropland
fires, it treats croplands as natural grasslands. The emission factors of grasslands and croplands used in the FireMIP
models are larger than in GFED4s, thereby causing larger fire $PM_{2.5}$ emissions in southeastern US in CLM and
JULES (van der Werf et al., 2017; Li et al., 2019). Furthermore, Li et al. (2019) noted that CLM4.5 simulates
higher fuel loads in croplands than the CASA model used by GFED4s, leading to higher fire carbon emissions
estimated by CLM than by GFED. It is worth noting that the ML model incorporates fuel load simulated by CLM4.5
but it predicts fire emissions closer to GFED4s, indicating a smaller sensitivity of fire emission to fuel load in the
ML model. The overestimation of fire $PM_{2.5}$ emissions can also be explained by the sensitivity to meteorology. The
spatial distributions of the long-term mean fire emissions shown in Fig. 3 correlate with the spatial distributions of
sensitivities to RH and/or temperature, with regional differences. For western US, large fire emissions are
associated with stronger sensitivities to both RH and temperature in the ML and most FireMIP models. For central
and southeastern US, overestimation of fire $PM_{2.5}$ emissions only corresponds to stronger sensitivity to temperature
in some FireMIP models.
Besides comparing model performance aggregated over CONUS, we analyze the model performance for
several regions, including the western forest area, Mediterranean California, southwestern US, and southeastern
US. For the western forest area, the ML model performs well in capturing both seasonality and interannual
variability of fire $PM_{2.5}$ emissions, with correlation coefficients of 0.98 and 0.96, respectively. In contrast, the
FireMIP models generally reproduce the seasonality well but do not simulate the interannual variability well,
especially underestimating the peak in 2007, which related to large-scale meteorological patterns favorable for fires
in Pacific Northwest (Zhong et al., 2020). For Mediterranean California, all FireMIP models only capture the first
peak in August but fail to simulate the second peak in October, which is caused by large fires related to Santa Ana
winds in 2003 and 2007. Such lack of peak emission is also shown in the interannual variability, as all FireMIP
models show limited ability to simulate the peaks in these two years. By contrast, the ML model successfully
predicts the bimodal seasonality and the large fire emissions related to the Santa Ana winds in 2003 and 2007. The
underestimation of the peak in the FireMIP models may be attributed to the underrepresentation of the effects of
large-scale meteorology in the two regions, as the ML model and SHAP importance show that SVD predictors
have larger contributions to the fire emissions in both events. The results of the two regions in the western US
suggest that fire parameterization in the FireMIP models could be improved by including the effects of regional
and large-scale meteorology (e.g., Santa Ana winds) on fire activity (Yue et al., 2014). Modeling the effect of Santa
Ana winds on wildfires may be particularly challenging as the offshore Santa Ana winds exhibit variability related
to both synoptic scale pressure anomaly over the Great Basin and local thermodynamic forcing associated with
strong desert-ocean temperature gradient (Hughes and Hall, 2010).
As for southwestern US, the ML model and LPJ-SIM estimate the peak in June (r=0.99 and 0.94 for ML
and LPJ-SIM, respectively), which highly agrees with the GFED observation. Interestingly, most FireMIP models



fail to capture the extremely large fire emission in the 2011 summer mainly due to the low biases of VPD anomalies in CRUNCEP (Tang et a., 2017). Unlike southwestern US, the seasonality of southeastern US has peaks in March-April and September-October. The two peaks of fire emissions correspond to wildfires (Mar-Apr and Sep), cropland fires (Feb-Mar and Aug-Oct), and prescribed fires (Feb-Apr and Oct) that include burnings for pest controls and land cleaning (Knapp et al., 2009; Lin et al., 2014). Most models fail to reproduce the bimodal fire emissions, but the ML model, LPJ-SIM, and LPJ-SPI can capture the bimodal pattern. Even though the seasonality of fires over this region is not simulated accurately, the CLM and JSBACH well reproduce the interannual variability of fire $PM_{2.5}$ emissions and predict the peaks. The FireMIP models' shortfall in reproducing the bimodal seasonality can be explained by two reasons. First, the relationships between human and fire spread implemented in the process-based models may not be realistic compared to the observed relationships. Parisien et al. (2016) demonstrated the large spatial variability of human impacts on burned areas in North America, which is not well represented in the FireMIP models (Li et al., 2019). Second, drier conditions in winter would promote fires in springtime (Wear and Greis, 2013; Wang et al., 2021), which may not be directly considered in the FireMIP models but are incorporated as SVD predictors in the ML model. Overall, the representations of the effects of human and large-scale meteorology on fires may explain why the models simulate the seasonality incorrectly in southeastern US. In addition to the comparison of general model performance, we also compare the ability of the data-driven and processed-based models in predicting extremely large fire emissions. Both ML and CLM models reproduce the spatial pattern of extreme fire events and reasonably simulate the historical events of large fires in southwestern and western US.

To summarize, we utilize the ML model with SHAP importance to diagnose the fire emissions simulated by process-based models and attributed model biases to several factors. First, the sensitivities of fire emissions to meteorology in the models are stronger than the observed, leading to overestimations. Second, the large-scale meteorological patterns conducive to fires are not fully considered in the process-based models, which are important contributors of large fire emissions in western US and southeastern US. Third, the spatial resolutions of models and/or the atmospheric forcing they used may be too coarse to resolve the effects of regional weather phenomenon such as Santa Ana winds. Fourth, biases in the atmospheric forcing data may result in biases of fire emission predictions. Last but not least, human activities are a critical component shaping fire regimes but the human effects on fire activities in the FireMIP models may not reflect the human-fire relationships in the real-world. This is also an issue in the ML model as the human-related predictors in the ML model may be too simple to represent the human influences. The underrepresentation of human effects in both types of models may cause additional uncertainties in projecting future fire activities and their impacts on climate. By training the ML model using the GFED emissions, the ML model is able to better explain fire emissions in the US, which makes it a useful tool for diagnosing processes or relationships that may be missing or not well represented in the process-based models to guide future development for improving their performance. Besides its use in diagnosing process-based models, the interpretable ML model provides a different and novel approach to simulate fire emissions more accurately and identify the important predictor variables. While the ML model generally has higher accuracy than the FireMIP models, the feedbacks between fire emissions and climate are not included, which could potentially affect the reliability of ML-based models in fire emission prediction under future climate change scenario (Zou et al., 2020). Lastly, due to limited training data, the ML model cannot predict fires in regions with longer fire return intervals, posing additional uncertainties in their use for making future projections.

*Code availability.* Model code is available upon request to the first author.

*Data availability.* The ML prediction and predictor dataset used in this study are publicly accessible online at https://zenodo.org/record/5076646#.YOZI4zZKjOQ.






*Author contributions.* SW, YQ, RL conceived the research ideas. SW wrote the initial draft of the paper, performed
analyses, and model development. All authors contributed to the interpretation of the results and the preparation of
the manuscript.

*Acknowledgements.*
This research was performed at PNNL and funded under Assistance Agreement No. RD835871 by the U.S.
Environmental Protection Agency to Yale University through the SEARCH (Solutions for Energy, AiR, Climate,
and Health) Center. It has not been formally reviewed by EPA. The views expressed in this document are solely
those of the SEARCH Center and do not necessarily reflect those of the Agency. EPA does not endorse any products
or commercial services mentioned in this publication.

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

**Table 1.** Advantages and limitations of different types of fire models

|  | Representative method | Advantages | Limitations |
| --- | --- | --- | --- |
| **Data-driven model** | Multiple Linear regression (MLR) | 1. Computationally efficient<br>2. Simple model<br>3. It is easy to interpret | 1. It cannot capture the non-linear relationships between fires and predictors<br>2. It assumes that the predictor variables are independent<br>3. It is sensitive to outliers |
|  | Machine learning method (e.g., neural network, decision tree etc.) | 1. Computationally cheap<br>2. The performance improves when more training data are included<br>3. It can easily handle multi-dimensional data | 1. It requires a lot of training data<br>2. It is relatively hard to interpret<br>3. The interactions between fires and vegetation/atmosphere cannot be updated to the model |
| **Process-based model** | Dynamic global vegetation model (DGVM) | 1. Physics-driven<br>2. The simulations can include feedbacks between fires and climate or vegetation | 1. Computationally expensive<br>2. The same parameterization may not be applied to all regions<br>3. It only parameterizes the known processes or phenomena |






**Table 2.** Predictor variables used in the ML model

| Variables | Abbreviation | Categories | Temporal resolution | Spatial resolution | Data Source | References |
|---|---|---|---|---|---|---|
| Monthly mean surface temperature | temp | Local meteorology | monthly | 32 km | North American Reanalysis (NARR) | Mesinger et al. (2006) |
| Monthly mean relative humidity | RH | Local meteorology | monthly | 32 km | North American Reanalysis (NARR) | Mesinger et al. (2006) |
| Monthly mean of daily precipitation | precip | Local meteorology | monthly | 32 km | North American Reanalysis (NARR) | Mesinger et al. (2006) |
| Monthly mean zonal component of wind speed | U | Local meteorology | monthly | 32 km | North American Reanalysis (NARR) | Mesinger et al. (2006) |
| Monthly mean meridional component of wind speed | V | Local meteorology | monthly | 32 km | North American Reanalysis (NARR) | Mesinger et al. (2006) |
| Monthly Standardized Precipitation Evapotranspiration Index | SPEI | Local meteorology | monthly | 0.5°×0.5° | SPEI | Vicente-Serrano et al. (2010) |
| Monthly mean 1000-hour dead fuel moisture | FM1000 | Local meteorology | daily | 4 km | gridMET | Abatzoglou (2013) |
| Monthly mean energy release component | ERC | Local meteorology | daily | 4 km | gridMET | Abatzoglou (2013) |
| Monthly mean vapor pressure deficit | VPD | Local meteorology | daily | 4 km | gridMET | Abatzoglou (2013) |
| Monthly lightning flashes density | lightning | Local meteorology | daily | 0.1°×0.1° | SWDI/NLDN | NOAA (2006); Cummins and Murphy (2009) |
| Monthly standard deviation of daily SVDs for northern California | SVD1_NCA and SVD2_NCA | Large-scale meteorological patterns | monthly | Regional | North American Reanalysis (NARR) | Wang et al. (2021) |



| | | | | | | |
|---|---|---|---|---|---|---|
| Monthly standard deviation of daily SVDs for southern Rocky Mountain | SVD1_SRM and SVD2_SRM | Large-scale meteorological patterns | monthly | Regional | North American Reanalysis (NARR) | Wang et al. (2021) |
| Monthly standard deviation of daily SVDs for southeastern US (with 2-month lag) | SVD1_SElag2 and SVD2_SElag2 | Large-scale meteorological patterns | monthly | Regional | North American Reanalysis (NARR) | Wang et al. (2021) |
| Monthly mean evapotranspiration | ET | Land-surface properties | monthly | 0.25º×0.25º | North American Land Data Assimilation System (NLDAS-2) | Xia et al. (2012) |
| Monthly mean surface soil moisture | soilm | Land-surface properties | monthly | 0.25º×0.25º | Global Land Data Assimilation System (GLDAS-2) | Xia et al. (2012) |
| Monthly mean vegetation fraction | Veg_frac | Land-surface properties | monthly | 0.25º×0.25º | Global Land Data Assimilation System (GLDAS-2) | Xia et al. (2012) |
| Monthly mean Leaf Area Index | LAI | Land-surface properties | 8 days | 500 m | MODerate resolution Imaging Spectroradiometer (MODIS); LAI classification scheme | Myneni et al. (2015) |
| Monthly fuel load/normalized fuel load | fuel_load/fuel_load_nor | Land-surface properties | monthly | 0.9º×1.25º | Community Land Model (CLM) | Lawrence et al. (2019) |
| Land cover percentage | p_ | Land-surface properties | Yearly | 0.05º×0.05º | MODerate resolution Imaging Spectroradiometer (MODIS); LAI classification scheme | Friedl (2015) |
| Median Topography (slope and elevation) | Slope and elevation | Land-surface properties | Not change by time | 100 km | | Amatulli et al. (2018) |
| Gross domestic product | GDP | Socioeconomic and coordinate variables | Yearly | 5 arc | | Kummu et al. (2018) |
| Population density | Pop | Socioeconomic and coordinate variables | Yearly | 30 arc | Gridded Population of the World data collection (GPW v4) | CIESIN-Columbia University (2017) |





**Table 3.** The ML model performance for different regions: western forest area, Mediterranean California, southwestern US,
and southeastern US

| | Western forest area | Mediterranean California | Southwestern US | Southeastern US | Whole US |
|---|---|---|---|---|---|
| **Grid scale (individual grid)** | | | | | |
| RMSE (km$^2$) | 0.29 | 0.32 | 0.10 | 0.02 | 0.16 |
| Correlation (r) | 0.79 | 0.51 | 0.76 | 0.84 | 0.75 |
| IoA | 0.86 | 0.60 | 0.85 | 0.90 | 0.84 |
| Percentage of grids with correlation > 0.4 (%) | 68 | 47 | 52 | 80 | 74 |
| **Regional scale (summation over the region)** | | | | | |
| RMSE (km$^2$) | 37.80 | 13.94 | 2.76 | 3.37 | 49.98 |
| Correlation (r) | 0.98 | 0.81 | 0.94 | 0.97 | 0.97 |
| IoA | 0.98 | 0.81 | 0.95 | 0.98 | 0.97 |



**Table 4.** The model performance for the ML model and FireMIP models

| | ML model | CLM | CTEM | JSBACH | LPJ-SPI | LPJ-Glob | LPJ-SIM | ORCHIDEE | JULES |
|---|---|---|---|---|---|---|---|---|---|
| **Total amounts of fire PM2.5 emissions (Tg=10$^{12}$ g) (GFED: 4.89 Tg)** | | | | | | | | | |
| Total fire PM2.5 emissions over 2000-2012 (Tg) | 8.33 | 16.54 | 41.50 | 19.92 | 16.23 | 79.49 | 35.38 | 2.43 | 33.43 |
| **Correlation of interannual/seasonal variability for the CONUS** | | | | | | | | | |
| Correlation (interannual/seasonal) | 0.87/0.98 | 0.71/0.92 | 0.28/0.87 | 0.15/0.89 | 0.15/0.92 | 0.02/- | 0.23/0.65 | 0.03/0.91 | 0.55/0.93 |
| **Correlation of interannual/seasonal variability for the selected regions** | | | | | | | | | |
| Western forest area | 0.93/0.98 | 0.70/0.93 | 0.33/0.88 | 0.21/0.88 | 0.38/0.79 | 0.51/- | 0.46/0.98 | 0.05/0.94 | 0.60/0.92 |
| Mediterranean California | 0.72/0.97 | -0.01/0.87 | 0.05/0.94 | -0.30/0.89 | -0.07/0.90 | -0.14/- | -0.19/0.83 | 0.25/0.13 | -0.21/0.87 |





| | | | | | | | | | |
|---|---|---|---|---|---|---|---|---|---|
| Southwestern US | 0.95/0.99 | 0.14/0.85 | -0.26/0.62 | -0.28/0.45 | 0.34/0.42 | 0.30/- | 0.40/0.94 | 0.45/0.72 | -0.07/0.69 |
| Southeastern US | 0.96/0.99 | 0.57/-0.27 | -0.16/0.09 | 0.72/-0.14 | 0.08/0.35 | 0.39/- | 0.18/0.68 | 0.16/0.13 | 0.36/0.01 |



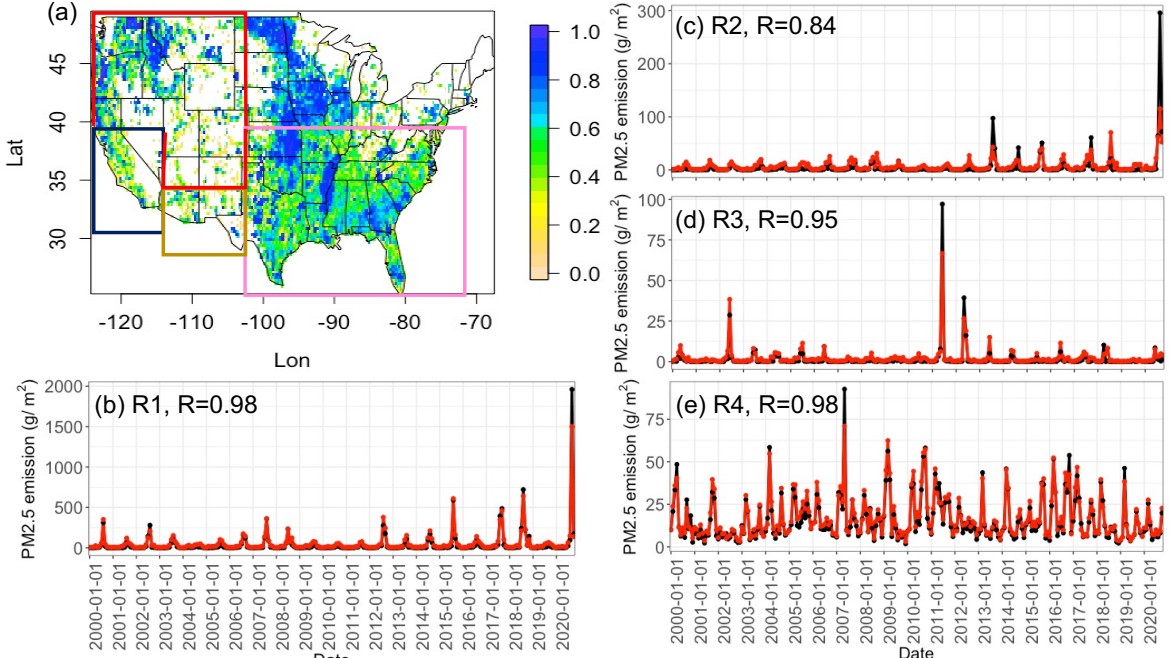


**Fig. 1.** (a) The map of temporal correlation between observed and predicted PM₂.₅ fire emission for each grid. Time series of observed (black) and predicted (red) PM₂.₅ fire emission average across (b) western forest area (red box in 1a), (c) Mediterranean California (blue box), (d) southwestern US (dusty box), (e) southeastern US (pink box).

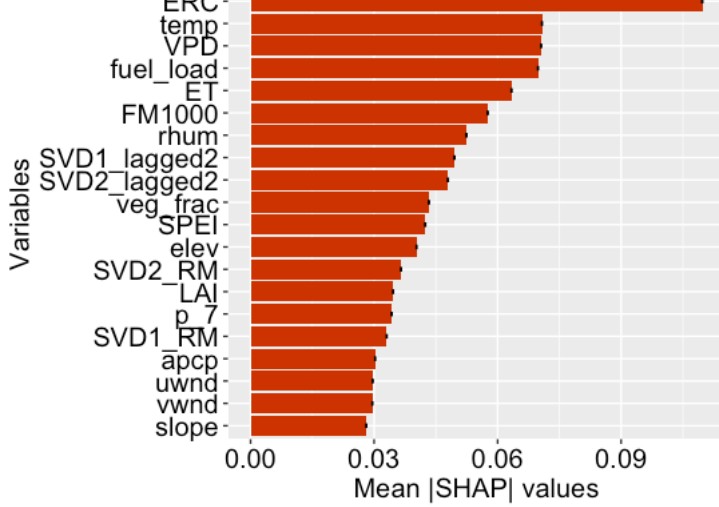






**Fig. 2.** Top 20 variables for the model based on the mean absolute SHAP value with the 95% confidence intervals.

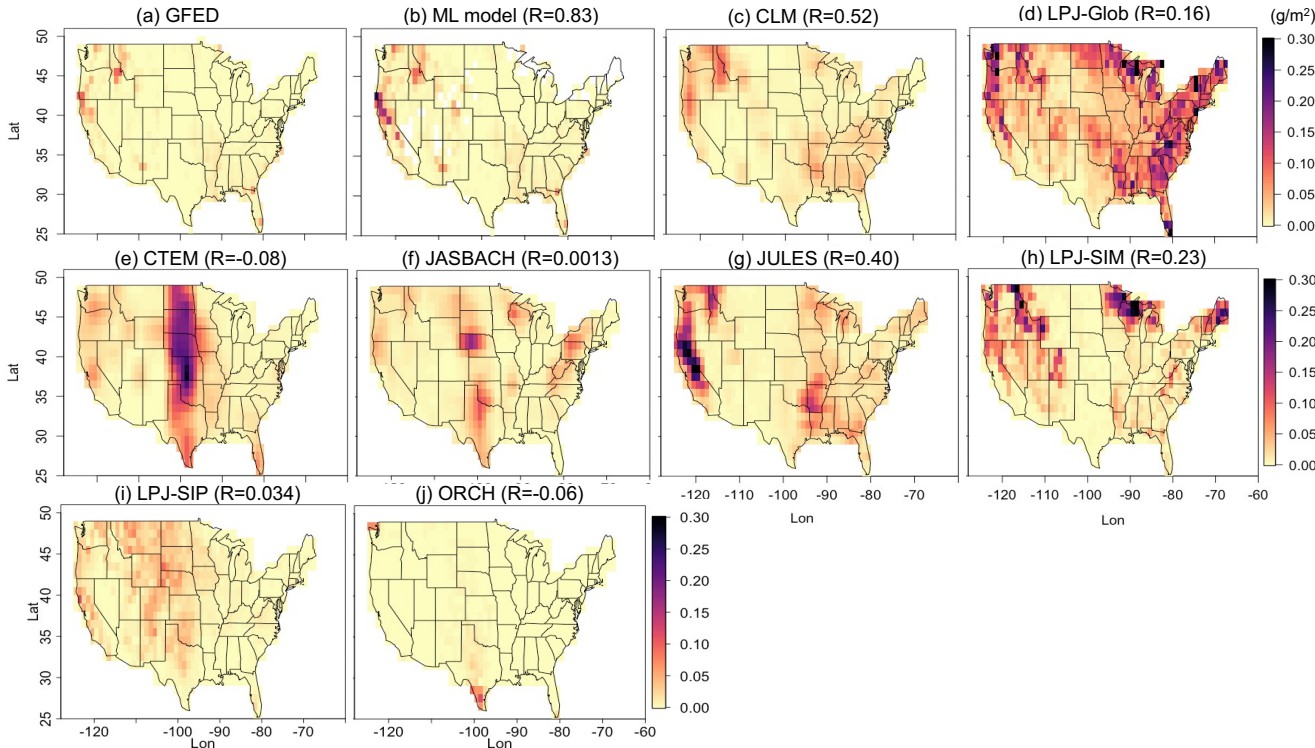


**Fig. 3.** Spatial distributions of the monthly mean PM$_{2.5}$ fire emission (g/m$^2$/month) over 2000-2012.



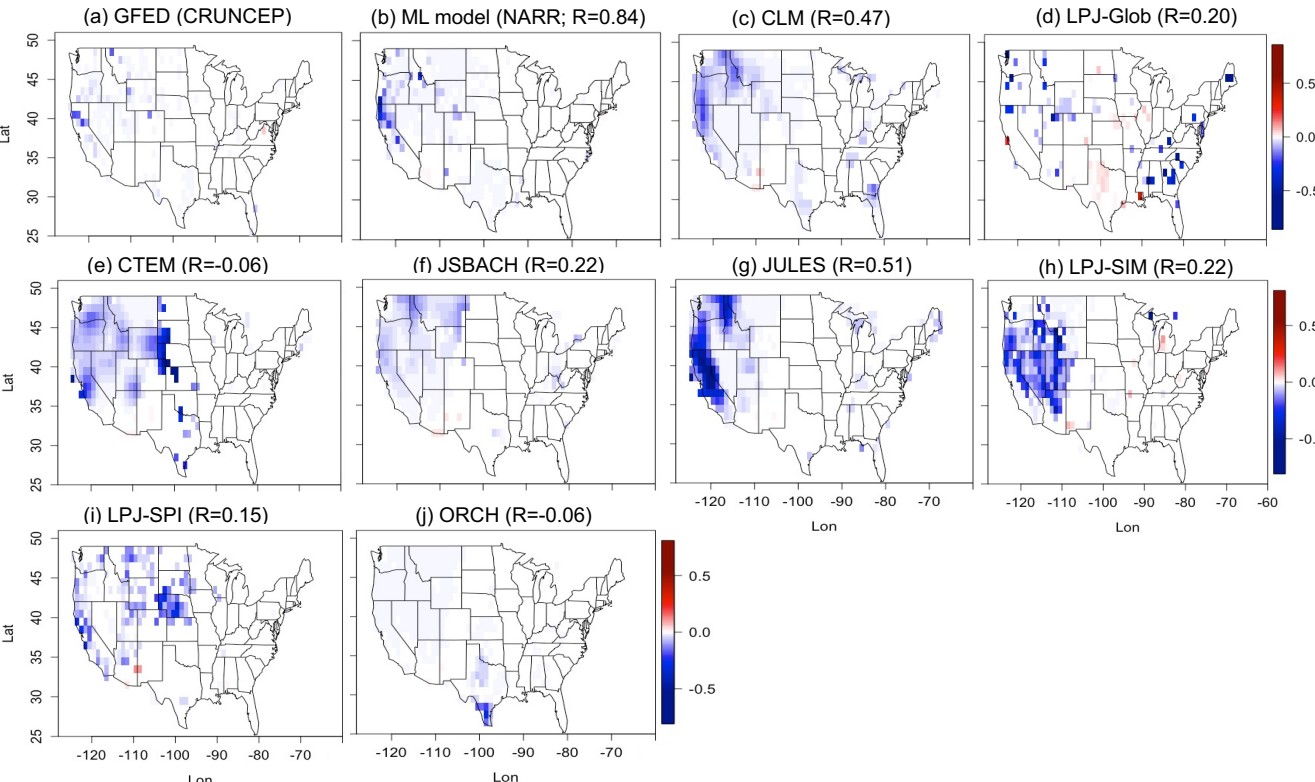

**Fig. 4.** Spatial distributions of the linear regression slope for the dependence of annual mean $PM_{2.5}$ fire emissions on annual mean RH. Only the grids with slopes that are statistically significant ($p<0.05$) are shown.





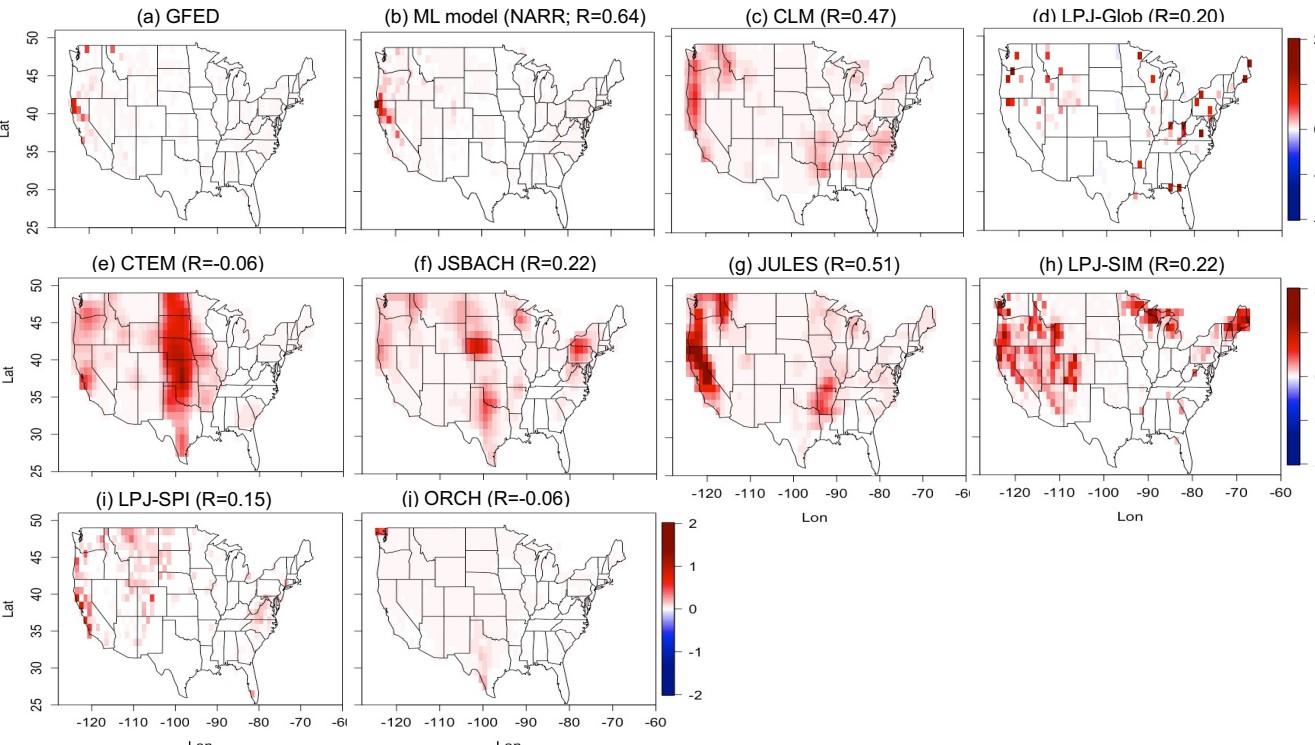

**Fig. 5.** Spatial distributions of the linear regression slope for the dependence of annual mean PM$_{2.5}$ fire emissions on annual mean temperature. Only the grids with slopes that are statistically significant (p<0.05) are shown.





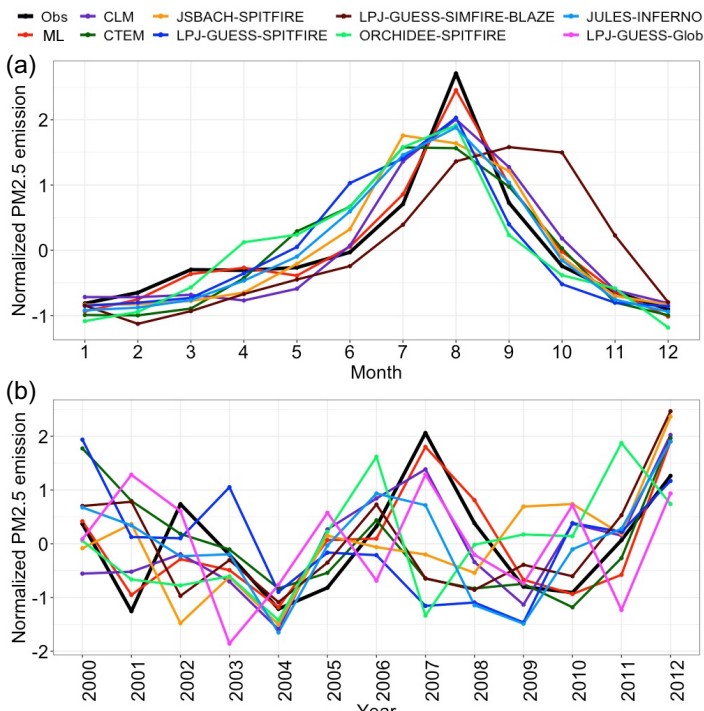

872

**Fig. 6.** (a) Seasonality and (b) interannual variability of the normalized averaged $PM_{2.5}$ fire emission from the GFED (black line), ML model (red line), and the FireMIP models (color lines). The $PM_{2.5}$ fire emissions are first averaged over CONUS and normalized by the monthly (annual) mean and standard deviation for seasonality (interannual variability) plots.

876





**Fig. 7.** Seasonality and interannual variability of the PM2.5 fire emission from the GFED (black line), ML model (red line), and the FireMIP models (color lines) for (a, b) western forest area, (c, d) Mediterranean California, (e, f) southwestern US, and (g, h) southeastern US.



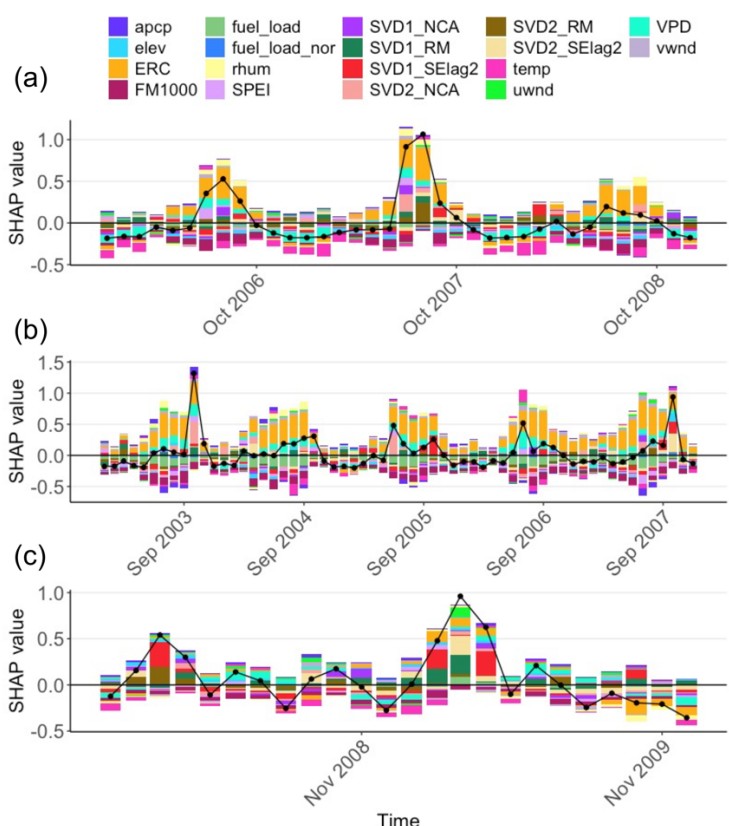


**Fig. 8.** Time series of the average SHAP values (bar) and predicted normalized PM$_{2.5}$ fire emission (line) for (a) western forest area from 2006 to 2008, (b) Mediterranean California from 2003 to 2007, and (c) southeastern US from 2008 to 2009. The SHAP values indicate the contribution of the predictors to the prediction of normalized fire emission.



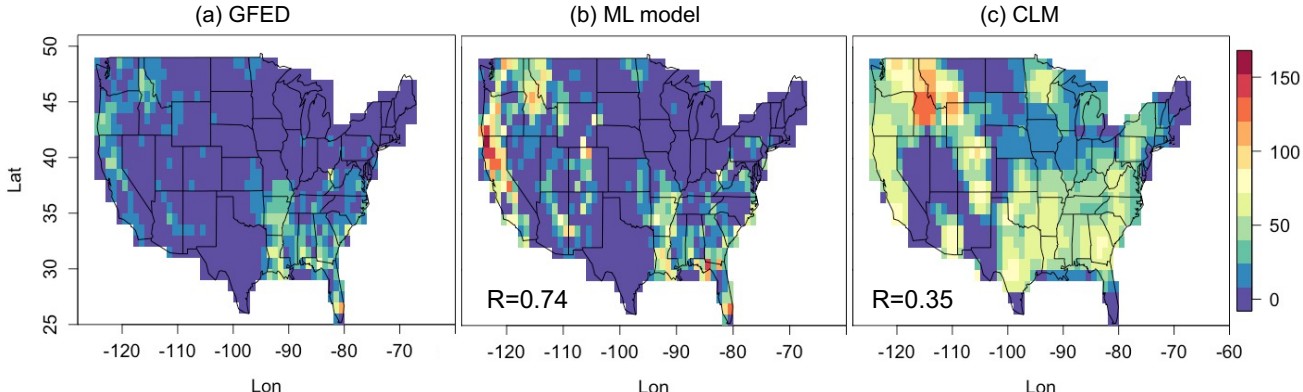


**Fig. 9.** Spatial distributions of number of months with large fire emissions (> 95[th] percentiles of PM$_{2.5}$ fire emission over all the grids in 2000-2012) for (a) GFED, (b) ML model, and (c) CLM.


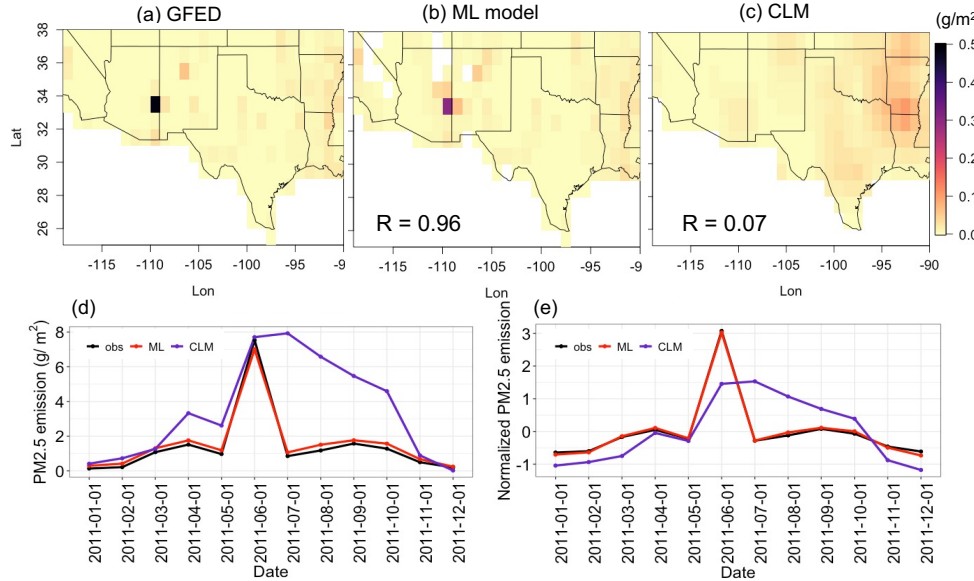


**Fig. 10.** Top panel: Spatial distributions of the annual mean PM$_{2.5}$ fire emission in 2011 for (a) GFED, (b) ML model, and (c) CLM. Bottom panel: Time series of the (d) total PM$_{2.5}$ fire emissions and (e) normalized PM$_{2.5}$ fire emission over southern US domain during 2011.


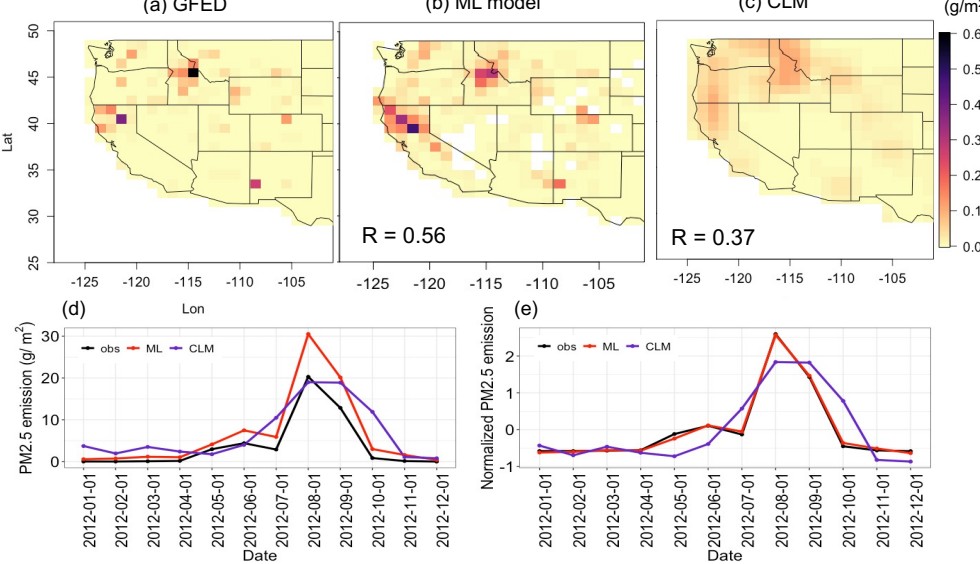


**Fig. 11.** Top panel: Spatial distributions of the annual mean PM$_{2.5}$ fire emission in 2012 for (a) GFED, (b) ML model, and (c) CLM. Bottom panel: Time series of the (d) total PM$_{2.5}$ fire emissions and (e) normalized PM$_{2.5}$ fire emission over western US domain during 2012.