# Peer review of "Interpreting machine learning prediction of fire emissions and comparison with FireMIP process-based models"

_Atmospheric Chemistry and Physics, 2021_

## Author Comment (AC1)

**Response to Reviews**

We thank the two reviewers for their constructive comments to improve the manuscript. Their comments are copied below with our responses shown in blue. The corresponding changes in the manuscript are also highlighted in blue and the associated changes in the revised manuscript are marked in green.

**Reviewer #1**

1. Interpretable ML or interpreting ML

There is a major difference in internally interpretable ML models versus trying to interpret ML outcomes with analytic tools [Rudin 2019]. I think this study belongs to the second category.

In that case, the title is really confusing, first XGBoost model is explainable that one can easily derive variable importance. However, SHAP was finally used to interpret XBGoost.

Interpreting an interpretable ML model (XGBoost) with an interpreting tool (SHAP) is a little bit weird. Why not just used XGBoost derived variable importance? What's the value of SHAP? If SHAP is really necessary and could give us a better understanding of XGBoost, then at least the title needs to be updated to: e.g., "Interpreting ML prediction of fire emission xxx". And focus more on why SHAP is better than XGBoost internal variable ranking.

Thank you for your comments. We use SHAP instead of taking the variable importance directly from XGBoost because SHAP provides local importance, which offers variable importance for each sample. In contrast, XGBoost only provides global importance that evaluates the variable importance of all the samples. The local importance helps understand which variables have larger contributions to a specific event, which cannot be assessed using the variable importance from the XGBoost model, as stated in Section 3.3 (lines 268-273). The SHAP value for a feature indicates its contribution to the prediction. We have stressed the reason for choosing SHAP to understand how the XGBoost model learns from the data to predict fire emissions in the manuscript, as shown below:

**Lines 304-309:**

SHAP importance is chosen because it provides not only global importance but also local importance that helps understand which variables have larger contributions to specific events or regions. Here, we first demonstrate the global importance that considers all the samples. Fig. 2 shows the 20 most important variables for the model ranked by the absolute mean SHAP values. The SHAP value for a feature indicates its contribution to the prediction, so larger absolute mean SHAP values indicate larger contributions to the fire emissions.

To more accurately describe this study, we have replaced "interpretable ML" with "ML model with SHAP" in the main text and revised the title to "Interpreting machine learning prediction of fire emissions and comparison with FireMIP process-based models".

2. What can we learn and to inform future development

As highlighted in the abstract that one of the objectives was to inform model future development. However, it wasn't sufficiently discussed and there is no clear conclusion on e.g., which part of the process-based model needs major development? What is missed in process-based models?

Based on the results, we identified several factors that contributed to the biases in the processbased models and summarized them in the last paragraph of Section 5 – Discussion and conclusions. These factors include: (1) process-based models have larger sensitivities of fire emissions to meteorology compared to the sensitivities in observations; (2) the large-scale meteorological patterns conducive to fires are not fully considered; (3) the spatial resolutions of the models might be too coarse to resolve the effects of regional weather patterns; (4) biases in the atmospheric forcing, and (5) human effects on fires might not be fully represented. These five aspects require additional developments for the process-based model to achieve better performance. See lines 611-631: "To summarize, we utilize the ML model …"

3. Different time scales: long-term trend, inter-annual variability, sub-seasonal dynamics

For different time scale, one would expect different dominant driver for wildfire burn and PM2.5 emissions. For example at the sub-seasonal scale, climate may play a more important role, while the long-term trends may be more affected by human activities. I wonder is it possible to carry out some experiments to better interpret the ML outputs across different scales? For example, detrend the time series (long-term trend) and use ML to predict interannual variability and compare the variable importance with the original ML models?

We agree that the dominant controlling factors might differ by time scales. To address the reviewer's concern, we aggregate the monthly SHAP values to obtain annual and seasonal time series of SHAP values for each variable. The annual and seasonal time series are the averaged SHAP values over the study period for each year and month, respectively. For each variable, we calculate the mean |SHAP| values and correlations between the annual/seasonal time series of variable SHAP and mean fire emissions. Larger mean |SHAP| values indicate larger contributions of the variables to the fire emissions, and higher correlations indicate that the variable contributions can better explain the variation of the fire emissions.

Fig. R1 shows the mean |SHAP| values at seasonal and interannual time scale for the whole CONUS. Considering both the mean |SHAP| and correlations larger than 0.5, temperature, VPD, RH, and ERC are the dominant variables controlling the seasonal variation of fire emissions. These factors have relatively stronger seasonality than other variables (e.g., VPD is usually higher in the summer). On the other hand, large-scale circulation patterns, including SVD1\_SElag2, SVD2\_SElag2, and SVD1\_RM, are important variables controlling both the seasonal and interannual variability of fire emissions, while SVD2\_RM and SVD2\_NCA mainly control interannual variability. Some identified large-scale meteorology has significant seasonality (e.g., SVDs\_SElag2 are predominant in spring and SVD1\_RM is strongest in summer), and most of them have interannual variability (i.e., their contributions vary by years), as shown in Fig. 8. Overall, the SHAP analysis shows different dominant predictors for fire emissions at various time scales.

**Fig. R1** Variable importance in mean |SHAP| values at seasonal and interannual time scale. The variables and time scales which have correlation between time series of fire emissions and SHAP values larger than 0.5 are in stripes.

We have included the results and associated discussions in the manuscript (lines 322-334) and Fig. R1 is also included in the supplement as Fig. S7.

**Lines 322-334:**

As the dominant drivers differ for different temporal scales, we aggregate the monthly SHAP values to obtain annual and seasonal time series of SHAP values for each variable. The annual and seasonal time series are the averaged SHAP values over the study period for each year and month, respectively. Fig. S7 shows the mean |SHAP| values at seasonal and interannual time scale for the whole CONUS. Considering both the mean |SHAP| and larger correlations (r > 0.5) between the annual/seasonal time series of SHAP and mean fire emissions, temperature, VPD, RH, and ERC are the dominant variables controlling the seasonal variation of fire emissions. These factors have relatively stronger seasonality than other variables (e.g., VPD is usually higher in the summer). On the other hand, large-scale circulation patterns, including SVD1\_SElag2, SVD2\_SElag2, and SVD1\_RM, are important variables controlling both the seasonal and interannual variability of fire emissions, while SVD2\_RM and SVD2\_NCA mainly control interannual variability. Some identified large-scale meteorology has significant seasonality (e.g., SVDs\_SElag2 are predominant in spring and SVD1\_RM is strongest in summer), and most of them have interannual variability, as shown in Fig. 8. Overall, the SHAP analysis shows different dominant predictors for fire emissions at various time scales.

**4. Uncertainties in data and ML model training/prediction**

There are multiple existing datasets (e.g., Fire Atlas, Fire CCI). One potential issue of training/validating an ML model using only GFED data is that the ML predictions are subject to GFED uncertainties. If possible, a comparison of GFED with other products and even better, applying GFED emission factors to other BA products, then one can train/validate ML model towards multiple datasets of fire emissions, include the data uncertainties in the cost function.

We acknowledge that other fire datasets are available, including Fire Atlas, Fire CCI, FINN, QFED, GFAS, etc. We chose GFED4s because (1) it is one of the most widely used global fire emission inventories, (2) it provides fire emission estimations, and (3) it has more extended temporal coverage than other datasets. Prior studies have compared the estimated fire emissions based on GFED and other products (Liu et al., 2020; Li et al., 2019). They noted that the GFED data tends to underestimate the fire emission peak in springtime over the southeastern US, which may be explained by the fact that other products such as FINN capture more small fire activity compared to the GFED approach (Koplitz et al., 2018; Carter et al., 2020). Despite this known discrepancy between GFED and other data products, the normalized PM2.5 emissions of GFED still show bimodal peaks in spring and fall over the southeastern US, while most FireMIP models fail to reproduce the first peak (Fig. 7g in the manuscript). We appreciate reviewer's suggestion of applying GFED emission factors to other BA products to quantify data uncertainties; however, large uncertainties exist in the emission factor and fuel consumption (Davis et al., 2015; Ottmar, 2014), which may lead to additional uncertainties when quantifying the data uncertainty.

In the revised manuscript, we have added discussions of uncertainties in fire emissions datasets, including findings from previous studies that compared GFED with other data products, as shown below. We also noted the need for future work to address data uncertainties by incorporating uncertainty information in the cost function used to train and validate ML models.

**Lines 596-610:**

It is known that different fire emission inventories have their uncertainties and prior studies have compared fire emission inventories over the globe or CONUS (Urbanski et al., 2018; Liu et al., 2020). The GFED fire emissions used in this study are known to underestimate the fire emission peak in springtime over the southeastern US, which may be explained by the fact that other products such as FINN or QFED capture more small fire activity compared to the GFED approach (Koplitz et al., 2018; Carter et al., 2020). Although FINN can capture more small fires, it underestimates the intensity of large fires for some cases, which has been attributed to the cloud coverage on daily scale detection (Paton-Walsh et al., 2012). QFED and GFAS, which estimate emissions using fire radiative power (FRP) from satellites, are also more sensitive to small fires than GFED. However, QFED tends to estimate much larger emissions than other products, which can be explained by the fact that the emission coefficients used to obtain emissions are constrained by MODIS AOD and the uncertainties within FRP (Pan et al., 2020). Despite the known discrepancy between GFED and other data products, the GFED data still shows bimodal peaks in spring and fall over the southeastern US, while most FireMIP models fail to reproduce the first peak (Fig. 7g in the manuscript). For the western US, GFED and FINN are generally consistent regarding the magnitude and variability of fire emissions (Urbanski et al., 2018). As stated above, different fire emission inventories have uncertainties. Future works are required to include other fire emission datasets for model evaluation.

**Others:**

5. L23: xxx, which may be explained by the coarse spatial resolutions of the processed-based models or atmospheric forcing data or limitations in model parameterizations for capturing the effects of Santa Ana winds on fire activity. This statement is not helpful. What is the real reason

why the FireMIP model did not capture bimodal peak emissions? For example, one can check wind fields in GSWP3 forcings or CRUNCEP forcings to verify the existence of Santa Ana winds.

To confirm our statement, we compare the wind speed between CRUNCEP and NARR, focusing on the days with strong wind speed (daily mean wind speed >4.5 m/s) over southwestern California (116-119 °W, 32.6-34.8 °N) during 2000-2012 October. In October, large wind speeds (>4.5 m/s) in southwestern California are usually offshore winds and are associated with Santa Ana winds (Figure 7 in Yue et al., 2014). The wind speeds from NARR are significantly larger than from CRUNCEP for the identified large wind speed days (Fig. R2). In addition, we also compare the mean wind speed of the large wind speed days for Oct 2003 and 2007, as shown in Table R1. For both years, the mean wind speeds from NARR are significantly stronger than CRUNCEP, indicating the lower wind speeds in the CRUNCEP used in FireMIP may partially explain the model biases for the events associated with Santa Ana winds (large wind speed).

**Fig. R2** Distribution of wind speed for the days with large wind speeds (>4.5 m/s) in October during 2000-2012 for the CRUNCEP and NARR data. The NARR data is regridded to  $0.5^{\circ}$  x 0.5°, matching the spatial resolution of the CRUNCEP data. This figure is now included in supplement as Fig. S8.

| Table R1. Mean wind speed of Oct 2003 and 2007 over southwestern California for the tw | 0 |
|----------------------------------------------------------------------------------------|---|
| atasets.                                                                               |   |

| Wind speed (m/s) | Oct 2003 | Oct 2007 | Oct mean |
|------------------|----------|----------|----------|
| CRUNCEP          | 3.75     | 4.98     | 4.41     |
| NARR             | 5.07     | 5.67     | 5.40     |

We have revised the sentence to explain the biases of the process-based models and included the results in the manuscript, as shown below:

**Lines 20-23:**

However, all models except for the ML model fail to reproduce the bimodal peaks in July and October over Mediterranean California, which may be explained by the smaller wind speeds of the atmospheric forcing data during Santa Ana wind events and limitations in model parameterizations for capturing the effects of Santa Ana winds on fire activity. Lines 436-440:

As shown in Fig. S8, the wind speeds from NARR are significantly larger than from CRUNCEP for the strong wind days (daily wind speed > 4.5 m/s) over southwestern California (116-119 °W, 32.6-34.8 °N) during 2000-2012 October as well as the during Oct 2003 and 2007 (Table S3). The results indicate the lower wind speeds in the CRUNCEP atmospheric forcing used in FireMIP may partially explain the model biases for the events associated with Santa Ana winds.

6. L108: How was emission factor data estimated? Are the emission factors PFT dependent or constant across the whole US?

The emission factors were obtained from Akagi et al. (2011). The emission factors are dependent on the fire types, including savanna, boreal forest, temperate forest, tropical forest, and agriculture (van der Werf et al. (2017). We included the abovementioned information in the manuscript, as shown below:

**Lines 112-117:**

The GFED fire PM2.5 emissions are estimated by combining the burned area boosted by small fire burned area (Randerson et al., 2012) and the emission factors based on Akagi et al. (2011) with a revised version of the Carnegie-Ames-Stanford Approach (CASA) biogeochemical model that estimates fuel loads and combustion completeness for each monthly time step (van der Werf et al., 2017). The emission factors are dependent on the fire types, including savanna, boreal forest, temperate forest, tropical forest, and agriculture (van der Werf et al., 2017).

7. Section 2.2. What are the differences in input variables used by FireMIP and ML model? For a fair comparison, it will be good to make sure ML and FireMIP models used the same input variables. But, if not, what are the implications of using different input variables, and how do they contribute to the ML and FireMIP model differences?

The ML model uses input variables that are primarily included as input variables in the processbased models. To compare the input variables between the ML and FireMIP models, we reviewed the literatures and summarized them in Table R2. As shown in Table R2, the ML and FireMIP models share many common variables, including precipitation, temperature, wind speed, relative humidity, lightning flashes density, and population density. Additionally, even among the FireMIP models, input variables are not the same. Since the purpose of comparing the fire emissions predicted by the ML model and the process-based models is to help diagnose predictions by the process-based models to inform future developments, we do not limit the ML model to using the same input variables as used in the process-based models, as shown below. For example, as we mentioned in Section 4.2.3, all FireMIP models fail to reproduce the second fire emission peak in October over Mediterranean California, while the ML model predicts the peak successfully. Given the differences in winds in the atmospheric forcing used in the ML model and process-based models (see our response to comment #5), and the differences in predictors (i.e., ML model includes large-scale meteorology as predictors while the FireMIP models do not have such input variables or related parameterizations), we can conclude that including predictors of large-scale meteorological patterns favorable for fires improves the ML model performance.

Lines 90-99:

It uses the XGBoost algorithm and incorporates various predictors, including local and largescale meteorology, land surface characteristics, and socioeconomic variables, which are common input variables also used by the FireMIP models while some are specifically related to fire activities in CONUS. We acknowledge that different input variables between the ML and FireMIP models might cause additional uncertainty for comparison. This study aims to construct an ML model that predicts fire emissions over CONUS and utilize the ML model and SHAP to reveal the important factors contributing to fire emissions that might not be fully represented in the process-based models. In this context, the ML model and FireMIP models are optimized using different data or predictors at various scales, which enables us to use the ML to diagnose the performance of FireMIP models over CONUS through the comparisons of their performances and variable importance from the ML model.

| Table R2. Input variables used to drive models (only the variables not calculated from the |
|--------------------------------------------------------------------------------------------|
| models are included). The common variables are marked in bold.                             |

|                                       | Meteorological input variables                                                                                                                                                    | Socioeconomic variables    | References                                         |
|---------------------------------------|-----------------------------------------------------------------------------------------------------------------------------------------------------------------------------------|----------------------------|----------------------------------------------------|
| ML model                              | Temperature, RH,
precipitation, wind speed
(U&V), SPEI, ERC, VPD,
lightning flashes density , SVDs
(large-scale meteorology)                     | Population density,
GPD | This study                                         |
| CLM                                   | Temperature, RH, specific
humidity, precipitation, wind
speed, air pressure, downward
solar radiation, lightning
frequency                                            | Population density         | Li et al. (2012); Li
et al. (2013)              |
| CTEM                                  | Temperature, precipitation,
specific humidity, surface
pressure, wind speed , shortwave
and longwave radiation,
lightning flashes , soil texture | Population density         | Arora & Boer
(2005); Melton and
Arora (2016) |
| LPJ-
SPITFIRE
(LPJ-SPI)         | Temperature, precipitation ,
number of wet days, cloudiness,
wind speed , atmospheric CO 2 ,
lightning flashes                           | Population density         | Thonicke et al.
(2010)                          |
| LPJ-
GUESS-
Glob (LPJ-
Glob) | Temperature, precipitation ,
percentage sunshine hours, soil
texture                                                                                                 |                            | Thonicke et al. (2001)                             |
| LPJ-
GUESS-                        | Temperature, precipitation , downward shortwave radiation                                                                                                                  | Population density         | Knorr et al. (2016)                                |

| SIMFIRE   |                                           |                    |                 |
|-----------|-------------------------------------------|--------------------|-----------------|
| (LPJ-SIM) |                                           |                    |                 |
| JSBACH-   | Temperature, precipitation,               | Population density | Thonicke et al. |
| SPITFIRE  | number of wet days, cloudiness,           |                    | (2010)          |
| (JSBACH)  | wind speed, atmospheric CO 2 , |                    |                 |
|           | lightning flashes                         |                    |                 |
| JULES-    | Temperature, RH,                          | Population density | Mangeon et al.  |
| INFERNO   | precipitation, lightning flashes,         |                    | (2016)          |
| (JULES)   | soil moisture                             |                    |                 |
| ORCHDE    | Temperature, precipitation,               | Population density | Thonicke et al. |
| E-        | number of wet days, cloudiness,           |                    | (2010)          |
| SPITFIRE  | wind speed, atmospheric CO 2 , |                    |                 |
| (ORCHID   | lightning flashes                         |                    |                 |
| EE)       | _                                         |                    |                 |

8. L171: LAI is a poor indicator of biomass since the majority of the biomass comes from the stem. As long as the canopy is closed, the growth of vegetation biomass no longer link to LAI.

LAI correlates with canopy bulk density that describes the density of available canopy fuel in a stand and is important for crown fires (Keane et al., 2005). We understand that LAI may not fully represent the available biomass, so we also include vegetation fraction and fuel load from CLM as predictors to represent various fuel characteristics for different types of fires. We have included the above discussions in the manuscript, as shown below:

**Lines 176-181:**

As there are limited observations of fuel load, we use LAI to approximate the canopy bulk density, which is important crown characteristics to predict crown fire spread, and vegetation fraction to represent the existing amount of vegetation (Keane et al., 2005). LAI is taken from MODerate resolution Imaging Spectroradiometer (MODIS) instruments (Myneni et al., 2015) and vegetation fraction is obtained from the NLDAS-2. As LAI may not fully represent the available biomass, we also include fuel load simulated by Community Land Model (CLM).

9. L173: Worth first evaluate CLM fuel load with existing present-day biomass datasets, then apply it to ML model.

We included the comparison between CLM fuel load and the field-measured fuel load from the global fuel consumption database (van der Werf et al., 2017; van Leeuwen et al., 2014). Figure R3 shows the scatter plot between the observed and CLM-simulated fuel load over CONUS. Based on the results, the simulated fuel loads from CLM are consistent with the measured fuel loads across cropland, temperate forest, and boreal forest (Fig. R3). Due to the limited measurements, there are only 15 observations available across CONUS, and more data is required for more comprehensive evaluations. We have included the results in the manuscript, as shown below:

Lines 184-187:

CLM fuel load is validated by comparing with the fuel-measured fuel load from the global fuel consumption database (van der Werf et al., 2017; Van Leeuwen et al., 2014), as shown in Fig. S4. The CLM-simulated fuel load is generally consistent with the measured fuel for different vegetation types across CONUS based on the limited measurements.

**Fig. R3** Scatter plot between the observed and CLM-simulated fuel load over CONUS. Three fire types are marked in pink (cropland), green (temperate forest), and black (boreal forest). The data is obtained from fuel consumption database (van der Werf et al., 2017; Van Leeuwen et al., 2014). This figure is now included in supplement as Fig. S4.

10. L202: FireMIP models used cru-ncep, while ML model used NARR and gridMET, ML's fuel load input was simulated with GSWP3 forcings? The differences in climate forcings make the comparison less valuable, especially when forcing uncertainties dominated the comparison. Maybe one can eliminate the forcing uncertainties by first surrogate FireMIP with ML models and replace CRUNCEP forcing with the GSWP3 forcings.

We chose the simulated fuel load driven by the GSWP3 instead of CRUNCEP forcing because we would like to have data in both the historical (2000-2014) and projection (2015-2020) runs from the same ensemble member (r270). The CLM output of historical run driven by CRUNCEP from the same ensemble does not include land use and land cover change, which is important for changes in spatial distributions of fires. Therefore, we chose GSWP3 output instead to ensure data consistency for the two time periods.

This study aims to construct an ML model that predicts fire emissions over CONUS and utilize the interpretable ML model (SHAP) to reveal the important factors contributing to fire emissions that might not be fully represented in the process-based models. In this context, this goal is better achieved by optimizing the data used to develop the ML model so that the ML model is useful for informing the process-based models. Indeed, by using the reanalysis data with finer resolutions (NARR), the ML model can better explain the fire emissions in CONUS, which enables us to diagnose processes or relationships that may be missing or not well represented in the process-based models. For instance, by comparing the input meteorology data of the ML and FireMIP models over the southwestern US in 2011, we attribute the low biases of FireMIP models to the biases in the atmospheric forcing (lines 468-472). Considering the goals of this study, developing surrogated models of the FireMIP models may lead to additional uncertainties that complicate diagnosis of the FireMIP model biases (e.g., whether the ML models can perfectly reproduce the FireMIP outputs). However, we acknowledge the uncertainties of different inputs and include the discussions in the manuscript, as shown below:

**Lines 93-99:**

We acknowledge that different input variables between the ML and FireMIP models might cause additional uncertainty for comparison. This study aims to construct an ML model that predicts fire emissions over CONUS and utilize the ML model and SHAP to reveal the important factors contributing to fire emissions that might not be fully represented in the process-based models. In this context, the ML model and FireMIP models are optimized using different data or predictors at various scales, which enables us to use the ML to diagnose the performance of FireMIP models over CONUS through the comparisons of their performances and variable importance from the ML model.

11. Section 2.3. Has the FireMIP model sufficiently tuned using GFED data? Since the ML model was maximally tuned towards the GFED dataset, it's important to clarify whether or not FireMIP also tuned towards GFED. Otherwise, it's expected that the ML model would outperform FireMIP models.

We carefully reviewed the literatures of the FireMIP models but could not find information of model tuning. Even though not all FireMIP models were tuned using GFED fire data, Li et al. (2019) showed that models that perform well against other satellite products usually have good agreements with GFED (e.g., CLM and CTEM). Here we again emphasize that the purpose of comparing the fire emissions predicted by the ML model and the FireMIP models is to help diagnose the FireMIP model biases. We have clarified the goals of this study in the abstract and Section 1 and included the discussions regarding the reviewer's concerns in Section 1 and 5.

Lines 12-13: The optimized ML model is used to diagnose the process-based models in the Fire Modeling Intercomparison Project (FireMIP) to inform future development.

**Lines 93-99:**

We acknowledge that different input variables between the ML and FireMIP models might cause additional uncertainty for comparison. This study aims to construct an ML model that predicts fire emissions over CONUS and utilize the ML model and SHAP to reveal the important factors contributing to fire emissions that might not be fully represented in the process-based models. In this context, the ML model and FireMIP models are optimized using different data or predictors at various scales, which enables us to use the ML to diagnose the performance of FireMIP models over CONUS through the comparisons of their performances and variable importance from the ML model.

Lines 596-610:

It is known that different fire emission inventories have their uncertainties and prior studies have compared fire emission inventories over the globe or CONUS (Urbanski et al., 2018; Liu et al., 2020). The GFED fire emissions used in this study are known to underestimate the fire emission peak in springtime over the southeastern US, which may be explained by the fact that other products such as FINN or OFED capture more small fire activity compared to the GFED approach (Koplitz et al., 2018; Carter et al., 2020). Although FINN can capture more small fires, it underestimates the intensity of large fires for some cases, which has been attributed to the cloud coverage on daily scale detection (Paton-Walsh et al., 2012). QFED and GFAS, which estimate emissions using fire radiative power (FRP) from satellites, are also more sensitive to small fires than GFED. However, QFED tends to estimate much larger emissions than other products, which can be explained by the fact that the emission coefficients used to obtain emissions are constrained by MODIS AOD and the uncertainties within FRP (Pan et al., 2020). Despite the known discrepancy between GFED and other data products, the GFED data still shows bimodal peaks in spring and fall over the southeastern US, while most FireMIP models fail to reproduce the first peak (Fig. 7g in the manuscript). For the western US, GFED and FINN are generally consistent regarding the magnitude and variability of fire emissions (Urbanski et al., 2018). As stated above, different fire emission inventories have uncertainties. Future works are required to include other fire emission datasets for model evaluation.

12. Figure 4,5. The results will more meaningful if the regression was done for only the peak fire months (or fire season), given that emissions only happened during fire season.

We conducted the regression for the annual mean fire emission and meteorology because the fire seasons vary by region. For instance, the fire season in the western US is from June to October, while in the southeastern US the fire season has two peaks, one from March to May and the other from September to October (Fig. 7 in the manuscript). The sensitivity calculated based on different fire seasons may not be compared directly since the ranges of temperature (or RH) vary by season and region. Additionally, the peak fire season may not stay the same every year (e.g., October is the peak month in Mediterranean California, mainly for 2003 and 2007). Therefore, defining a single peak fire month or season to calculate the sensitivity may not be suitable for examining the fire-meteorology relationship.

**Reference**

Rudin, C. (2019). Stop explaining black box machine learning models for high stakes decisions and use interpretable models instead. Nature Machine Intelligence, 1(5), 206-215.

**Reviewer #2**

In this paper, the authors followed their previous study (Wang et al., 2021) and used ML technique for predicting fire emissions using gridded GFED fire emission dataset (as target) and meteorological, land-surface properties, and socioeconomic variables (as predictors). The performance of ML is evaluated and compared against FireMIP process-based models, and is interpreted using SHAP. The paper is clear written, and the scientific findings presented are important and suitable for the journal of ACP. However, there are a few issues need be clarified

before the paper is considered for publication. Here the reviewer recommends a "resubmission after major revisions".

We appreciate the reviewer's encouraging comments and valuable feedbacks. The major comments regarding fire datasets, model training, and variable selection were addressed and detailed in the response below.

General comments:

1. It is well known that different fire emission datasets (i.e. GFED, QFED, FINN, etc.) predict biomass burning emissions with large discrepancies, for example, Figure 6 in Liu et al. (2020). Large uncertainties associated with GFED dataset are due to the accumulated errors from burned area, fuel type/condition, and burning condition/fire weather. It is reasonable that ML results agree well with GFED, because it is trained against GFED. However, when validating FireMIP, the authors may consider the other fire emission dataset and examine whether the correlation coefficients are different. I understand that this requires additional work. The authors can ignore this suggestion but add additional discussion in the conclusion.

Prior studies have compared the estimated fire emissions based on GFED and other products over the globe or CONUS (Urbanski et al., 2018; Liu et al., 2020). Compared to other fire emission inventories, it has been noted that the GFED data underestimates the fire emission peak in springtime over the southeastern US, which may be explained by the fact that other products such as FINN or QFED capture more small fire activity compared to the GFED approach (Koplitz et al., 2018; Carter et al., 2020). Although FINN can capture more small fires, it underestimates the intensity of large fires for some cases, which has been attributed to the cloud coverage on daily scale detection (Paton-Walsh et al., 2012). QFED and GFAS, which estimate emissions using fire radiative power (FRP) from satellites, are also more sensitive to small fires than GFED. However, QFED tends to have a much larger estimation of emissions than other products, which can be explained by (1) the emission coefficient used to obtain emissions are constrained by MODIS AOD, thus resulting in higher scaling factors and derived emissions, and (2) the uncertainties within FRP (Pan et al., 2020).

Despite the known discrepancy between GFED and other data products, the GFED data still shows bimodal peaks in spring and fall over the southeastern US, while most FireMIP models fail to reproduce the first peak (Fig. 7g in the manuscript). For the western US, GFED and FINN are generally consistent regarding the magnitude and variability of fire emissions (Urbanski et al., 2015). In the revised manuscript, we have added discussion of uncertainties in fire emissions datasets, including findings from previous studies that compared GFED with other data products. We also noted the need to include other fire emission datasets for model evaluation in the future.

Lines 596-610:

It is known that different fire emission inventories have their uncertainties and prior studies have compared fire emission inventories over the globe or CONUS (Urbanski et al., 2018; Liu et al., 2020). The GFED fire emissions used in this study are known to underestimate the fire emission peak in springtime over the southeastern US, which may be explained by the fact that other products such as FINN or QFED capture more small fire activity compared to the GFED approach (Koplitz et al., 2018; Carter et al., 2020). Although FINN can capture more small fires,

it underestimates the intensity of large fires for some cases, which has been attributed to the cloud coverage on daily scale detection (Paton-Walsh et al., 2012). QFED and GFAS, which estimate emissions using fire radiative power (FRP) from satellites, are also more sensitive to small fires than GFED. However, QFED tends to estimate much larger emissions than other products, which can be explained by the fact that the emission coefficients used to obtain emissions are constrained by MODIS AOD and the uncertainties within FRP (Pan et al., 2020). Despite the known discrepancy between GFED and other data products, the GFED data still shows bimodal peaks in spring and fall over the southeastern US, while most FireMIP models fail to reproduce the first peak (Fig. 7g in the manuscript). For the western US, GFED and FINN are generally consistent regarding the magnitude and variability of fire emissions (Urbanski et al., 2018). As stated above, different fire emission inventories have uncertainties. Future works are required to include other fire emission datasets for model evaluation.

2. The way the ML model is trained guarantee the better performance of ML technique in capturing the interannual variability of fire, because the 10-fold cross-validation/random sampling method is used. When randomly splitting the whole sampling pool into 10 groups, the interannual variability information is stored in the 9 groups that are used for training. I strongly suggest the authors to examine the performance of ML by using entire or two years' data for validation purpose only. For example, the authors can train the ML using data from 2000 to 2019, and perform the ML model to the data of 2020, and examine the performance of ML model (total PM2.5, spatial distribution, seasonal variability, etc.). I believe by doing so, it can be considered as fair comparison against FireMIP models. In addition, the method will be more suitable for the future prediction. For the case studies, aren't the data of these two cases are included in training ML, unless I miss the text.

To address the reviewer's concern, we conducted several tests to demonstrate the ML model performance. The first test uses data from 2000 to 2019 as a training set and data from 2020 as a testing set. As shown in Fig. R4, the ML model is able to reproduce the spatial patterns of fire emissions well (r=0.72) but underestimates the total emissions, especially the peak in Sep 2020. The results are within our expectations because the ML model generally fails to make accurate predictions for the data outside of the training domain or has large uncertainties in extrapolation (Hooker, 2004; Tsubaki and Mizoguchi, 2020). Since 2020 features the largest fire emissions in the study period, we conducted another test using 2000-2017 and 2019-2020 to train the ML model and test on the data of 2018. We selected 2018 because 2018 had the largest fires on record before 2020. The ML successfully reproduces the total amount of the fire emissions in 2018, the temporal variability of fire emissions (r=0.92) and captures the peak in Aug 2018, as well as the spatial distributions of fire emissions (r=0.52).